# Temporal and spatial evolution of bottom-water hypoxia in the Estuary and Gulf of St. Lawrence

Mathilde Jutras[1,4], Alfonso Mucci[1,5], Gwenaëlle Chaillou[2,4], William A. Nesbitt[3] and Douglas W.R. Wallace[3]

[1]Department of Earth and Planetary Sciences, McGill University, 3450 University Street, Montreal, QC, H3A OE8, Canada
[2]Institut des Sciences de la Mer de Rimouski (ISMER) - Université du Québec à Rimouski, 300 Allée des Ursulines, Rimouski, QC, G5L 3A1, Canada
[3]Department of Oceanography, Dalhousie University, Steele Ocean Sciences Building, 1355 Oxford St., PO Box 15000, Halifax, NS, B3H 4R2, Canada
[4]Québec-Océan
[5]GEOTOP

*Correspondence to*: Mathilde Jutras (mathilde.jutras@mail.mcgill.ca)

**Abstract.** Persistent hypoxic bottom waters have developed in the Lower St. Lawrence Estuary (LSLE) and have impacted fish and benthic species distributions. Minimum dissolved oxygen concentrations decreased from ~125 µmol L$^{-1}$ (38% saturation) in the 1930s to ~65 µmol L$^{-1}$ (21% saturation) in 1984. Dissolved oxygen concentrations remained at hypoxic levels (< 62.5 µM = 2 mg l$^{-1}$ or 20% saturation) between 1984 and 2019 but, in 2020, they suddenly decreased to ~35 µmol L$^{-1}$. Concurrently, bottom-water temperatures in the LSLE have increased progressively from ~3°C in the 1930's to nearly 7°C in 2021. The main driver of deoxygenation and warming in the bottom waters of the Gulf and St. Lawrence Estuary is a change in the circulation pattern in the western North Atlantic, more specifically a decrease in the relative contribution of younger, well-oxygenated and cold Labrador Current Waters to the waters of the Laurentian Channel, a deep valley that extends from the continental shelf edge, through Cabot Strait, the Gulf and to the head of the LSLE. Hence, the warmer, oxygen-depleted North Atlantic Central Waters carried by the Gulf Stream now make up nearly 100% of the waters entering the Laurentian Channel. The areal extent of the hypoxic zone in the LSLE has varied since 1993 when it was first estimated at 1300 km². In 2021, it reached 9400 km², extending well into the western Gulf of St. Lawrence. Severely hypoxic waters are now also found at the end of the two deep channels that branch out from the Laurentian Channel, namely the Esquiman and Anticosti Channels.

## 1 Introduction

Hypoxia and anoxia occur naturally in many coastal environments with restricted circulation, such as fjords and embayments, but hypoxia in more open coastal and estuarine areas appears to be on the rise due to anthropogenic nutrient loading and coastal eutrophication (e.g. Saanich Inlet in British Columbia, Bedford Basin in Nova Scotia, Chesapeake Bay in

Maryland, shelf region of the northern Gulf of Mexico, the Kattegat in the Baltic Sea, the Bengali Current in western Africa, and the coastal area of the Changjian River/Estuary in the East China Sea, Bindoff et al., 2019; Breitburg et al., 2018; Gilbert et al., 2010; Rabalais et al., 2010; Li et al., 2002). Where the water column is shallow or seasonally stratified, such areas are not hypoxic throughout the year; they are ventilated seasonally through fall and winter mixing events (e.g., Gulf of Mexico).

Persistent hypoxia in coastal and estuarine environments is less common but has been identified at a number of locations where the water column is permanently and strongly stratified, including the Lower St. Lawrence Estuary in Eastern Canada (Genovesi et al., 2011; Thibodeau et al., 2006; Gilbert et al., 2005). Gilbert et al. (2005) first reported the presence of severely hypoxic bottom waters ($[O_2] < 62.5$ μM = 2 mg L$^{-1}$ or 20% saturation) in the Lower St. Lawrence Estuary (LSLE). In 2003, the hypoxic zone in the LSLE covered an estimated 1300 km$^2$.

Oxygen depletion in the bottom waters of the LSLE and the development of persistent hypoxic levels has had a direct impact on marine fauna and fisheries, including fish growth and distribution (Brown-Vuillemin et al., 2022; Dupont-Prinet et al., 2013; Petersen 2010; Chabot and Claireaux, 2008; Chabot and Dutil, 1999; D'Amours, 1993) and northern shrimp viability (Dupont-Prinet et al., 2013). At hypoxic levels, the benthic community structure is believed to undergo significant modifications (Riedel et al., 1997; Levin, 2003; Belley et al., 2010; Audet et al., 2022) while catabolic reactions and diagenetic cycling of redox-sensitive elements in the sediments are altered (Riedel et al., 1999; Katsev et al., 2007; Lefort et al., 2012).

## 1.1 The St. Lawrence Estuary

By most definitions, the Gulf and St. Lawrence Estuary make up the largest estuarine system on Earth (Figure 1). The greater St. Lawrence System connects the Great Lakes to the Atlantic Ocean. With a drainage basin of approximately 1.32 million km$^2$, the St. Lawrence River channels the second largest freshwater discharge (~12,000 m$^3$ s$^{-1}$) on the North American continent, second only to that of the Mississippi. The St. Lawrence Estuary (SLE) begins at the landward limit of the salt-water intrusion at the eastern tip of Île d'Orléans (5 km downstream of Québec City) and stretches 400 km seaward to Pointe-des-Monts where it widens into the Gulf of St. Lawrence (GSL). Traditionally, the SLE is divided into two segments based on its bathymetry and hydrographic features. The Upper St. Lawrence Estuary (USLE) extends from Île d'Orléans to Tadoussac, near the mouth of the Saguenay Fjord. This segment is relatively narrow (2 to 24 km wide) and mostly shallow, with depths typically under 30 m. Consequently, while it displays strong lateral salinity gradients, the water column is only weakly stratified. In contrast, the LSLE is much wider (30 to 50 km) and deeper (~300 m), displays a smoother, less variable bottom topography, and is more strongly stratified.

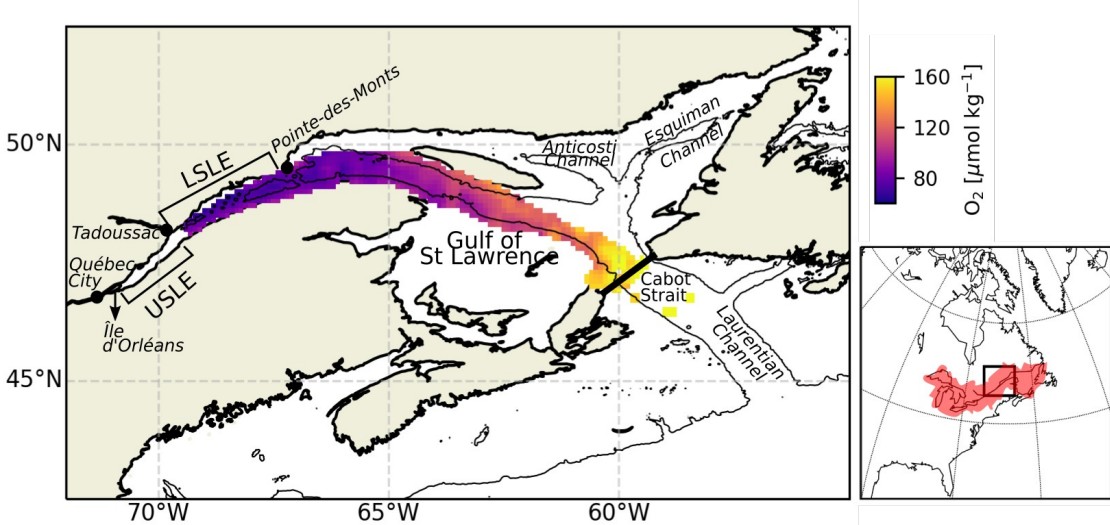

**Figure 1: Map of the LSLE, including the mean bottom-water oxygen concentration between 1970 and 2018. The thin black line delineates the 250 m isobath. The map on the right shows the drainage basin of the SLE.**

The dominant bathymetric feature of the Lower Estuary and Gulf is the Laurentian Channel (or Trough), a deep U-shaped submarine valley that extends 1280 km from the eastern Canadian continental shelf break through the GSL and into the LSLE. From this channel, two other deep channels branch out to the northeast within the gulf: the Esquiman and the Anticosti Channels (Figure 1). The LSLE is strongly stratified, and can be described as a three-layer system on the basis of its thermal stratification during the ice-free seasons. A warm and relatively fresh surface layer (0 to 30 m) that flows seaward overlies the

cold intermediate layer (CIL, 30-150 m deep; $S_P$ = 32.0 to 32.6, where $S_P$ stands for practical salinity) that flows landward and is formed in the wintertime in the GSL (Galbraith, 2006). Below the CIL, a warmer (2 to 7 °C) and saltier ($S_P$ = 33 to 35) bottom layer (> 150 m deep), isolated from the atmosphere, flows sluggishly landward (Dickie & Trites, 1983) for 4 to 7 years from the continental shelf-break to the head of the Laurentian Channel (Bugden et al., 1991; Gilbert, 2004). There, complex tidal phenomena due to rapid shoaling (tidal movements, including internal tides and strong flows over the steep sill) generate

significant local mixing of near-surface waters with deep nutrient-rich saline waters (Jutras et al., 2020a; Cyr et al., 2015; Saucier & Chassé, 2000; Ingram, 1983), resulting in a fertile surface layer that sustains a feeding habitat for several large marine mammals.

In this paper we provide an update on hypoxic conditions in the LSLE and GSL given the observed sudden drop in dissolved oxygen concentrations observed since our last report (Jutras et al., 2020b), and estimate the temporal evolution of the

areal extent of the hypoxic zone since our first report (Gilbert et al., 2005).

**2 Method**

We use in situ observations throughout the water column from three data sets. The first data set includes measurements we acquired mostly during the spring and summer between 2003 and 2021, and in the winder from 2018 to 2020, at multiple stations (Figure 2) along the Laurentian Channel, onboard the RV Alcide C. Horth, the R V Coriolis II and

the CCGS Amundsen. In all cases, sampling of the water column was carried out with a rosette system (12 or 24 x 12-L Niskin bottles) equipped with a Seabird 911Plus conductivity-temperature-depth (CTD) probe, a Seabird® SEB-43 oxygen

probe and a Seapoint® fluorometer. The Niskin bottles were closed at discrete depths as the rosette was raised from the bottom, typically at the surface (2-3m), 25m, 50m, 75m, 100m, and at 50m intervals to the bottom (or within 10m of the bottom). Even though the probes had been calibrated by the manufacturer within the year, discrete salinity samples were

collected throughout the water column and analyzed on a Guildline Autosal 8400 salinometer calibrated with IAPSO standard seawater and CTD profiles reprocessed post-cruise. Likewise, dissolved oxygen concentrations were determined by Winkler titration (Grasshoff et al., 1999) on discrete water samples recovered directly from the Niskin bottles. The relative standard deviation, based on replicate analyses of samples recovered from the same Niskin bottle, was better than 1 %. These measurements further served to calibrate the SBE-43 oxygen probe mounted on the rosette. The second data set was

extracted from the Bio-Chem database compiled by the Department of Fisheries and Oceans Canada. This data set provides quality-controlled data for the same variables as described above, and covers the Gulf and St. Lawrence Estuary from 1967 to 1972, and from 1991 to 2018, from spring to fall (Devine et al., 2014; DFO, 2019). A detailed description of data sampling techniques and quality control can be found in Devine et al. (2014) and Mitchell et al. (2002). Oxygen saturation levels are calculated from temperature, salinity and oxygen concentrations using the Python seawater package

(pypi.org/project/seawater/). The third data set contains oxygen concentrations measured by Winkler titration by clerics from Université Laval in the 1930s, and is used to extend the time series (Figure 4). We combine these three data sets, and aggregate the data per year, to focus on the inter-annual variability. The seasonality in air temperature and river runoff do not affect the bottom-water properties of the Laurentian Channel, as they are isolated from the surface by the intermediate layer (CIL). Only the spring bloom and the delivery of autochthonous organic matter could affect the bottom-water oxygen levels

on a seasonal scale, but available data show no consistent seasonality.

We reconstruct the causes of the 2018 to 2021 deoxygenation by applying an extended Optimum-Multiparameter (eOMP) analysis on the combined data set (Karstensen and Tomczak, 1998; Tomczak and Large, 1989; Tomczak, 1981). In this method, a set of linear equations that describe the properties ($S_P$, $\delta^{18}O(H_2O)$, temperature, alkalinity, dissolved oxygen and nutrient concentrations) of a parcel of water is used to determine the relative contributions of the different water types

that make up that parcel of water, given a definition of these water types in terms of the available water properties. Unlike the T-S diagram method, the eOMP method accounts for diapycnal mixing and provides estimates of biogeochemical changes that occurred between the water type formation and the measurement locations. Details of the application of this method to the current data set can be found in Jutras et al. (2020b).

The spatial extent of the hypoxic zone is estimated from dissolved oxygen (DO) concentration maps (Figure 2), by

computing the area included within the 275 m isobath from the head of the LSLE to the downstream-most measurement location displaying hypoxic dissolved oxygen concentrations. We aggregate all the measurements made during each year, and hence the estimates represent the maximal hypoxic area reached each year, during the sampled periods. Based on the available data, the spatial extent does not appear to vary seasonally. For the bathymetry, we use the General Bathymetric Chart of the Oceans (GEBCO) gridded bathymetry product with a 15 arc-second resolution. The uncertainty on the areal

estimate is calculated from the spatial extent between the seaward-most hypoxic station and the first non-hypoxic station. The area of hypoxic zones that are now observed at the head of the Esquiman and Anticosti Channels is not included in the calculation, given the large uncertainties at these locations due to lower data availability. This spatial extent calculation method has a number of limitations. First, even if the hypoxic waters often reach up to 250 m depth, we use a conservative value of 275 m for the isobath, as it is reached every year (Figure 3). This choice leads to an underestimation of the spatial

extent of the hypoxic zone in some years. Second, the real shape of the oxycline is likely domed close to the edges the Laurentian Channel, where it intersects the seafloor. There, turbulent mixing will dome the oxycline downward, while benthic respiration will dome it upward. The exact shape of the oxycline is not known, due to the weak spatial sampling rate in the St. Lawrence estuarine system. Yet, low resolution (every ~10 km) transects perpendicular to the Laurentian Channel

show a flat oxycline, suggesting that the doming is limited to the near edges of the channel. Hence, the error on estimates of the spatial extent of oxygen from an isobath will be small.

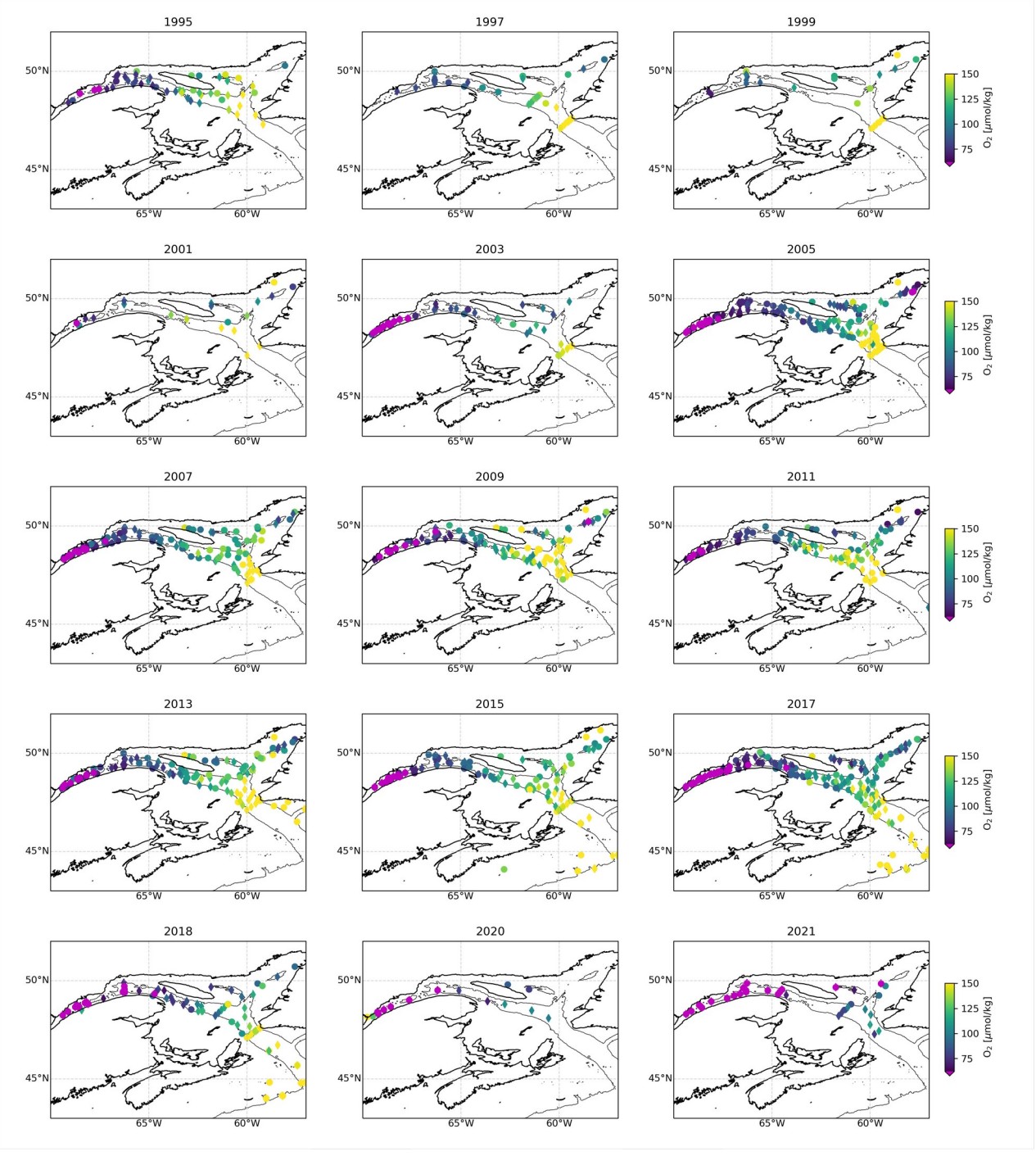

**Figure 2: Maps showing a compilation of all dissolved oxygen (DO) samples collected every year since 1995, for every station sampled deeper than 200 m. A map may contain data from multiple surveys carried out throughout that year. The color indicates the lowest DO concentration sampled over the water column, with hypoxic waters in magenta. The symbols indicate if the deepest**

**sample is located between 200 and 250 m (diamond) or below 250 m (circle). The thin black lines delineates the 275 m isobath.**

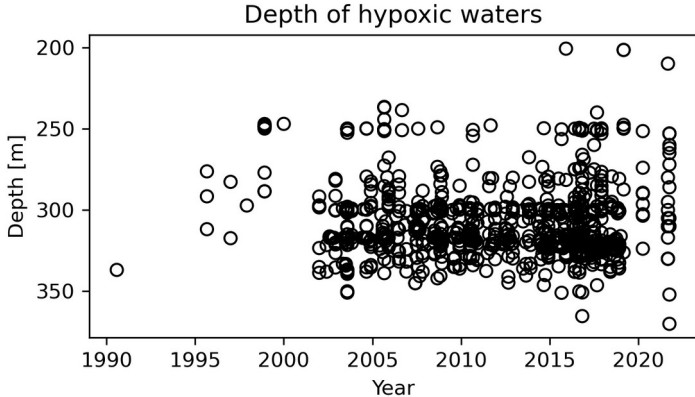

**Figure 3: Depth of hypoxic waters, through time. Each point represents one field observation of hypoxic waters (dissolved oxygen < 62.5 μM) within the LSLE.**


## 3 Results and discussion

### 3.1 Historical reconstruction

An historical reconstruction of published and unpublished field data clearly reveals that oxygen depletion in the bottom waters of the LSLE is a persistent feature of the system and is evolving in time (Figure 4 and 5, Jutras et al., 2020b;

Gilbert et al., 2005). The time series of bottom-water, minimum dissolved oxygen concentrations shows that, despite substantial inter-annual variability, three distinct clusters of points indicate that the range of yearly-averaged minimum bottom-water dissolved oxygen concentrations decreased from 110-135 μM in the 1930s to 95-120 μM in the early 1970s and then to 55-65 μM in the 1990s. A linear least squares fit applied to the dissolved oxygen time series between 1930 and 1985 yields a decreasing trend of about -1.0 ± 0.2 μM/year at the 95% confidence level. The situation seemed to have

stabilized after the mid-1980s as the trend in dissolved oxygen concentrations over the 1984-2016 period is not different from zero (-0.02 ± 0.79 μM/year at the 95% confidence level). In 2020, minimum dissolved oxygen concentrations suddenly decreased rapidly. Minimum dissolved oxygen levels were nearly cut by half within one year, reaching concentrations of ~35 μM, compared to 55-60 μM since early 2000s. Concurrently, bottom-water temperatures in the LSLE and the GSL have increased progressively from ~3°C in the 1930's to nearly 7°C in 2020, including a rapid 1°C increase from 2019 to 2020

(Figure 4). These levels are unprecedented in the recent history of the LSLE and the GSL, and impart an extremely high stress on marine life and the health of this ecosystem.

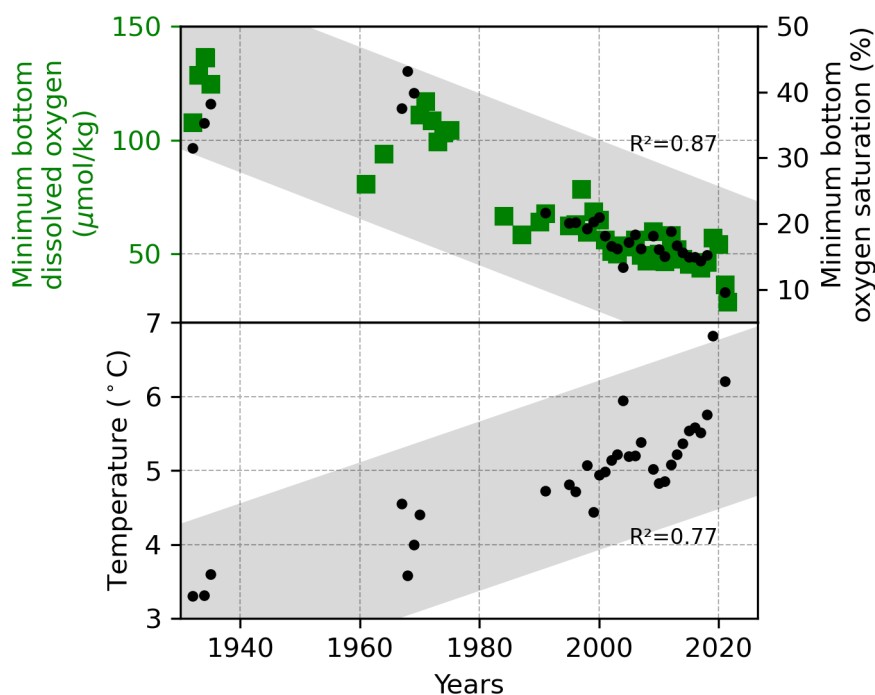

**Figure 4: Minimum bottom-water dissolved oxygen (DO) concentrations at the head of the LSLE (top), and associated water temperatures (bottom). The grey bands indicate the trend, based on a linear least-squares fit to the data, with the R² indicated on the figure. This figure offers a crude estimate of the trend in the bottom water properties, given that the exact sampling depth, although always below 250m, might have varied over the decades.**


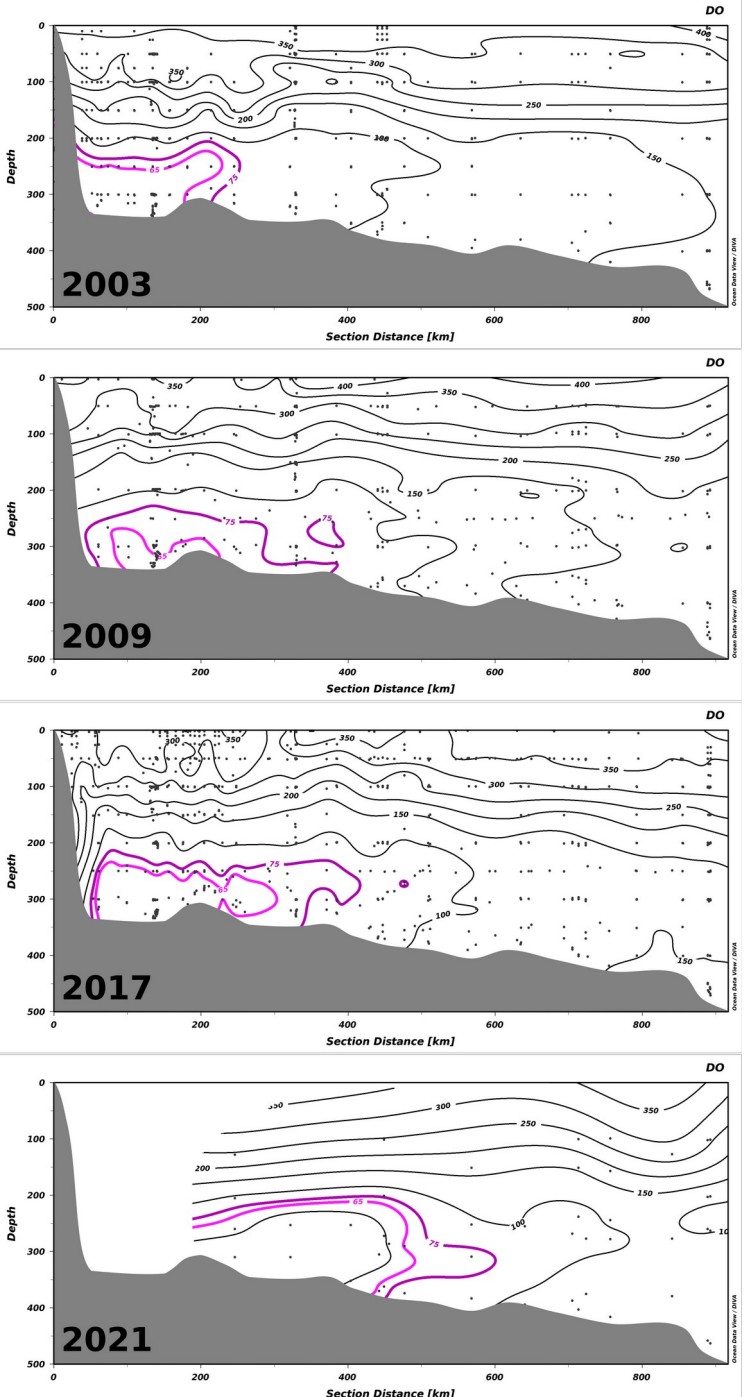

Figure 5: Transects of all dissolved oxygen (DO) concentration measurements along the Laurentian Channel, sampled in spring, summer and fall 2003, 2009, 2017 and in summer and fall 2021, from the head of the channel (left) to Cabot Strait (right). Black dots indicate the location of measurements. Contours show DO concentrations, in μM. The hypoxic waters are delineated with the magenta contour. Zones with no data are left blank.

## 3.2 Areal extent of the hypoxic zone

The progressive decline of bottom-water dissolved oxygen concentrations, including the most recent sudden decrease, is not only expressed in terms of a reduction in minimum dissolved oxygen concentration values, but also as an expansion of the hypoxic zone. Whereas it was estimated that the hypoxic zone covered 1300 km² in 2003 (Gilbert et al., 2005), it reached 9400 km² in 2021 (Figures 2, 5 and 6). The areal extent of the hypoxic zone has varied throughout the historical record, from a relatively stable 1300-2000 km² from 1995 to 2006, to 5000 km² in 2008-2011, concomitant with a
decrease in the relative proportion of LCW entering the Laurentian Channel and increased organic matter remineralization (Jutras et al., 2020b; Gilbert et al., 2005). The hypoxic zone then retreated to between 1000 and 2000 km² over the subsequent four years (2011-2015), before spreading to over 8000 km² in 2016. Contributing to this expansion, the area included within the 275 m isobath increases suddenly when the hypoxic zone reaches the Gulf of St Lawrence, where the Laurentian Channel widens. Notably, the five most spatially extensive hypoxic zones of the time series were recorded in the
last 5 years. Severely hypoxic waters are now also found at the end of the two deep channels that branch out from the Laurentian Channel, namely the Esquiman and Anticosti Channels (see years 2009 and 2021 in Figure 2).

In addition to the increasing spatial extent of the hypoxic zone, the thickness of the hypoxic layer is also increasing (Figure 3). Whereas the first observations of hypoxic waters were constrained to the bottom of the water column, between 275 and 350m depth (below the 27.25 sigma-tee isopycnal), the hypoxic layer now reaches deeper (as it expands spatially to
deeper sections of the Laurentian Channel) and shallower (up to 200 m, Figure 3). Overall, this increase in the areal extent and volume of hypoxic waters has strong consequences on this marine ecosystem, inducing stress and habitat compression.

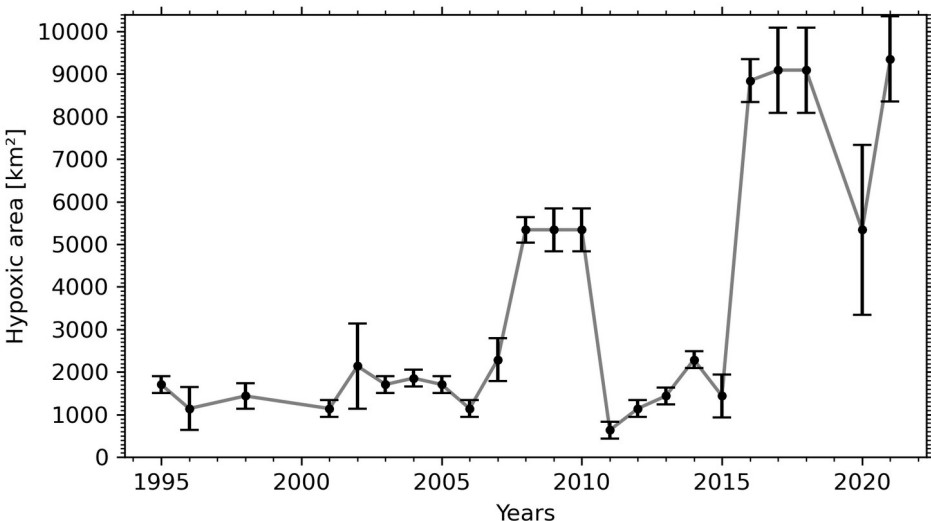

**Figure 6: Temporal variation of the areal extent of the hypoxic zone in the Laurentian Channel. The error bars indicate the uncertainty associated with each year's estimate, based on the spatial sampling rate (see section 2).**


## 3.3 Causes

The causes of deoxygenation in the Lower St. Lawrence Estuary from the 1930s to 2018 were discussed in detail elsewhere (e.g. Jutras et al., 2020b; Gilbert et al., 2005) and, thus, we here summarize their contribution to the most recent

decline of minimum dissolved oxygen (DO) concentrations (2018 to 2021). Several factors or combination of factors are responsible for the decrease observed over the last decades. Being isolated from the atmosphere, the bottom waters of the GSL and LSLE lose oxygen gradually through respiration and remineralization of organic matter as they flow landward (Figure 1). At depths greater than 150 m, the oxygen lost through microbial respiration in the water column and sediments cannot be replenished by winter convection, only by weak diffusion from the overlying water or by tidal mixing at the head of the Laurentian Channel (Cyr et al., 2015). For this reason, the oxygen balance is precarious: increased respiration and/or a decrease in deep estuarine landward flow will lead to lower dissolved oxygen concentrations. Unfortunately, there are few measurements of the mean flow of the bottom waters, but an analysis of the temperature field between 200 and 300 m depth along the Laurentian Channel reveals that the mean lateral flow may have increased slightly between two consecutive 26-year periods (1952-1977 and 1978-2003). Hence, in contrast to field observations, this should have resulted in an increase of the dissolved oxygen concentrations (Gilbert et al., 2004).

Analyses of the physical and biogeochemical properties of the deep waters of the LSLE since the early 1930s revealed that a change in water circulation in the western North Atlantic, more specifically the relative contributions of the two parent water masses (Labrador Current and North Atlantic Central Waters, LCW and NACW respectively) that mix on the continental shelf or slope and enter the Gulf of St. Lawrence (GSL) at depth through Cabot Strait, is responsible for most of the observed dissolved oxygen depletion and temperature increase (Jutras at al., 2020b; Gilbert et al., 2005). The LCW flow south along the western continental shelf edge of the Labrador Sea and then westward around the Grand Banks of Newfoundland, while the NACW are carried by the Gulf Stream. The former are colder and carry more DO than the latter (-0.7°C to 3.2°C and 310 to 280 µmol/kg of DO for the LCW, compared to 4.4°C to 17.7°C and 155 to 250 µmol/kg of DO for the NACW; Jutras et al., 2020b). Hence, if the proportion of Labrador Current water in the mixture feeding the bottom waters of the Laurentian Channel decreases, their temperature and salinity rise, and dissolved oxygen concentrations fall. Notwithstanding, the temporal variability of bottom-water DO concentrations does not exactly track the temperature and salinity variability, because eutrophication (Jutras et al., 2020b; Thibodeau et al., 2006) and an increase of the microbial respiration rates of settling organic matter in response to the increase in bottom-water temperatures (Genovesi et al., 2010) explains much of the remaining oxygen decline. The former is fostered by an increase in organic matter and nutrient exports from the estuary's main tributary, the St. Lawrence River, which drains a $1.32 \times 10^6$ km$^2$ basin that includes highly populated (>45 million Canadians and Americans; ECCC, 2018) and industrialized areas, as well as intensively farmed and forested lands (Hudon et al., 2017). Whereas a multi-decadal time series of nutrient discharge from the river to the estuary is not available, a record of fertilizer sales within the drainage basin is. The sales of phosphorus-based fertilizers grew rapidly in Quebec and Ontario starting in the 1940s and peaked in the late 1980s, after which they decreased slightly in the 1990's and came back to 1980 levels throughout the new millennium. The sales of nitrogen-based fertilizers increased steadily from the 1940s to 1980, stabilized throughout the 1980's and 1990's, but have since continued to increase significantly (Statistics Canada, 2018). Goyette et al. (2016) estimated that the net anthropogenic P and N inputs to watersheds of the St. Lawrence Basin have increased by respectively 3.8 and 4.5-fold over the last century.

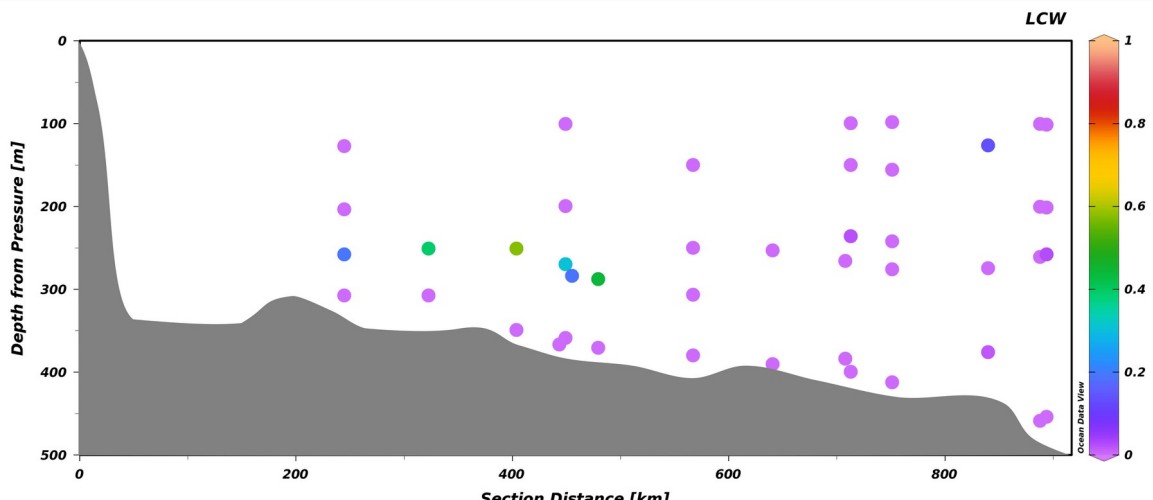

**Figure 7: Transect of the LCW fraction along the Laurentian Channel from a survey conducted between October 23 and 29, 2021, from the head of the Laurentian Channel (left) to Cabot Strait (right). The slightly offset points depict stations located along transects perpendicular to the channel.**

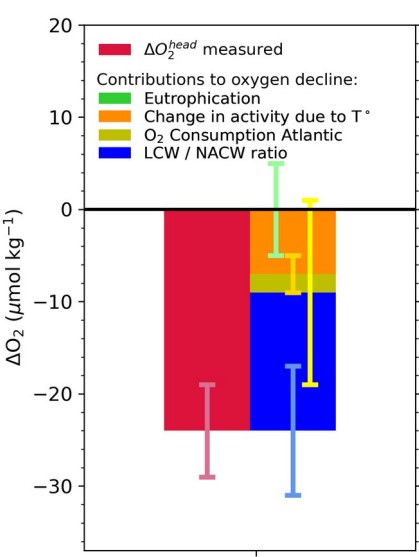

**Figure 8: Budget of drivers of deoxygenation at the head of the LSLE, from 2018 to 2021. Green: eutrophication within the Laurentian Channel; Orange: increased dissolved oxygen (DO) consumption within the channel under increased water temperature; Yellow: change in DO consumption in the North Atlantic, from the site of LCW and NACW mixing on the continental shelf or slope to Cabot Strait; Blue: modification of the LCW:NACW ratio at the entrance of the Laurentian Channel. See Jutras et al. (2020b) for details on the calculation method.**

Here, we apply the eOMP method used in Jutras et al. (2020b) (Section 2) to reconstruct the causes of the 2019 to 2021 oxygen decline (Figure 8). We refer the reader to the original paper for details on the properties of the different water masses, their distribution along the Laurentian Channel, and the temporal variability in their relative contributions to the

bottom waters of the Laurentian Channel. The analysis reveals that the proportion of LCW entering the Laurentian Channel is now null (rightmost portion of Figure 7), within the uncertainty (5%) of the eOMP method (see Jutras et al., 2020b). In other words, North-central Atlantic waters carried by the Gulf Stream now make up nearly 100% of the waters entering the Laurentian Channel. As the waters require 4 to 7 years to transit from the continental shelf break to the head of the Laurentian Channel, the presence of 20-50% LCW in the bottom waters close to the head of the channel (leftmost to central portion of Figure 7) reflects the proportion of LCW that entered the channel several years ago. The presence of residual LCW at the head of the LC suggests that bottom-water dissolved oxygen (DO) concentrations will further decline when the nearly undiluted oxygen-poor NACW reach the head of the LC. The increased NACW contribution explains more than 60% (~15 μM) of the deoxygenation (Figure 8) as well as the higher temperatures observed between 2018 and 2021. Note that this analysis only considers the warming due to a change in the mixing ratio, and does not consider temperature increases in the parent water masses themselves. Another 30% of the 2018 to 2021 deoxygenation is attributed to the increased biological oxygen consumption rates during the transit of waters along the Laurentian Channel in response to the temperature increase (Figure 8), estimated from the empirical temperature dependence of bacterial respiration rate $Q_{10}$ (Jutras et al., 2020b; Genovesi et al., 2011; Bailey & Ollis, 1986). The remaining 10% decrease is due to an increased dissolved oxygen consumption in the North Atlantic, also likely due to higher water temperatures, as estimated from results of the eOMP analysis at the entrance of the Laurentian Channel. In other words, the recent and sudden drop in bottom-water dissolved oxygen concentrations recorded since 2018 in the LSLE and GSL is entirely due to changes of the circulation patterns in the western North Atlantic, which affects dissolved oxygen both directly and indirectly, through the increase of water temperature. The recent increase in the amount of Gulf Stream water in the Slope Sea and feeding the LC is believed to be due to a slow-down of the Atlantic Meridional Overturning Circulation (AMOC, New et al., 2020), as well as to an increased retroflection of the Labrador Current towards the east in the vicinity of the Grand Banks in response to a stronger Labrador Current (Jutras et al., 2020b) and strong westerly winds over the Labrador Shelf (Holliday et al., 2020), both of which are related to a large-scale adjustment of the circulation patterns in the western North Atlantic (Jutras et al., in revision).

## 4 Conclusions

Recent observations revealed that minimum dissolved oxygen (DO) concentrations in the deep waters of the Lower St. Lawrence Estuary (LSLE) have reached unprecedented low values, dropping drastically in one year, from ~60μM or ~17% saturation in 2019 to ~35μM or ~10% in 2020. Concomitant with this sudden decrease in dissolved oxygen concentrations, bottom-water temperatures increased by one degree over the same period. Like for most of the recent historical record in the region, this deoxygenation is driven by a change in the circulation pattern in the western North Atlantic and in the relative contribution of the parent water masses (LCW and NACW), that mix on the continental shelf or slope, and flow along the axis of the Laurentian Channel. Based on a multi-parameter water mass analysis, we have determined that the contribution of the LCW to the mixture is now nearly null. The dissolved oxygen concentrations at the head of the estuary are expected to drop further, as these nearly pure NACW reach the head of the Laurentian Channel within the next 2 to 3 years. In fact, during our most recent survey (September, 2022) we detected waters with dissolved oxygen concentrations of less than 27 μM in the bottom waters of the LSLE.

Since the presence of hypoxic bottom waters in the LSLE was first reported in 2003, the areal extent of the hypoxic zone has increased from an estimated 1300 km$^2$ to more than 9400 km$^2$ in 2021, and now extends well into the western Gulf of St. Lawrence near the tip of the Gaspé Peninsula. In some years, patches of severely hypoxic bottom waters can also now be found at the tip of the Anticosti and Esquiman Channels where, however, historical data coverage is limited. This increase

in the areal extent of the hypoxic zone imposes a high stress on marine ecosystems, including unprecedented habitat compression.

## Data Availability Statement

The two main data sets for this research are the BioChem database compiled by the Department of Fisheries and Oceans Canada as well as data collected from the RV Coriolis (spring and summer) and CCGS Amundsen (winter) data set compiled by Alfonso Mucci's research group. The BioChem database can be accessed at [https://www.dfo-mpo.gc.ca/science/data-donnees/biochem/index-eng.html](https://www.dfo-mpo.gc.ca/science/data-donnees/biochem/index-eng.html) and the RV Coriolis and CCGS Amundsen data sets can be accessed at on the Open Science Framework data repository at osf.io/576tj/ and cited using the following DOI: 10.17605/OSF.IO/576TJ. For 2021,
we also include data collected during the MEOPAR TReX program and the OXY21 program. These data have been collected recently and are currently not publicly available.

## Author contribution

AM contributed to the conceptualization, funding, and writing. DW, AM, WN, MJ and GC contributed to the data acquisition, and DW, AM and GC to their curation, including providing the required resources. MJ contributed to the analysis,
visualization and writing.

## Acknowledgments

This study was funded by a Regroupement Stratégique grant from the Fonds de Recherche du Québec – Nature et Technologies (FRQNT) to GEOTOP, by the Natural Sciences and Engineering Research Council of Canada (NSERC) through Discovery grants to A. Mucci, G. Chaillou and D. Wallace, as well as support from MEOPAR and Réseau Québec
Maritime to D. Wallace for TReX (Tracer Release Experiment). M. Jutras acknowledges NSERC, the FQRNT, and Ouranos for financial support in the form of scholarships. We would like to acknowledge the help of the many students and technicians who contributed in sampling and the analysis of dissolved oxygen samples throughout the years, namely Alexandre Hérard, Joannie Cool, Olivier Hérard and Constance Guignard.

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
