# Peer review of "Temporal and spatial evolution of bottom-water hypoxia in the Estuary and Gulf of St. Lawrence"

_EGUsphere, 2022_

## Author Comment (AC1)

**Response to reviewer #1 on manuscript egusphere-2022-1090 titled "Temporal and spatial evolution of bottom-water hypoxia in the Estuary and Gulf of St. Lawrence" Comments posted Nov 10, 2022**

**RC1: This manuscript evaluates the temporal changes in hypoxia in the St. Lawrence River Estuary and Gulf from the earliest measurements in the 1930s to present. This manuscript is apparently an update of a previous paper by the main author (Jutras et al. 2020, JGR) with the main conclusion that hypoxia expanded drastically in recent years reaching almost 10,000 km2 in 2021. This conclusion may not be wrong but the authors have not managed to provide compelling evidence to support it!**

We thank the reviewer for his(her) incisive and helpful comments.

**RC1: First, the method used for assessing the extent of hypoxia is very coarse. The authors determine the area deeper than 275 m from the mouth of the St. Lawrence Channel to the farthest station with hypoxia. This approach does not consider that oxygen profiles are irregularly distributed along the gradient, which in combination with the gradual broadening of the channel can give quite variable results. The authors do acknowledge this limitation, but instead they should try to improve the spatial integration of the profiles by developing a more sound data processing approach, e.g. by fitting the oxycline as function of depth along the channel gradient. Moreover, hypoxic conditions can be observed at shallower depths than 275 m (cf. Fig. 3) and it can also be deeper, so why make this simplistic approach of considering the area deeper than 275 m. How do the authors define hypoxic conditions when oxygen concentrations are changing with depth? Importantly, more precise estimates can be obtained by minor improvements in the data processing through formulating an appropriate model.**

We agree with the reviewer that the method used to estimate the areal extent of hypoxia is rather coarse. Below, we explain why using the 275 m isobath is currently the most precise way to estimate the area of the hypoxic zone. The real shape of the hypoxic zone does not follow exactly the 275 m isobath, especially close to the edges of the Laurentian Channel, where turbulent mixing and benthic activity might dome the oxybaths. This possible vertical shift in the oxybaths, where the waters intercept the seafloor, is ont of the uncertainties in our method. However, the spatial sampling is too sparse to assess these effects.

- Mixing is expected to dome the hypoxic zone upward, while benthic oxygen consumption is expected to dome the hypoxic zone downward. Consequently, the two effects partially compensate each other, and we do not expect the overestimation or underestimation to be very significant.
- At the western limit of the hypoxic zone, at the head of the Laurentian Channel, we expect mixing to bring oxygen deeper in the water column, leading to a slight overestimation of the hypoxic area when using the 275 m isobath. However, the head of the channel represents a small fraction of the hypoxic region.
- Another point is that, while the hypoxic layer usually reaches up to 250 m (see Figures 3 and 5), we use the 275 m isobath, which is a safer choice that might lead to an underestimation of the hypoxic zone. We could clarify this point by changing the sentence "The 275 m isobath is chosen because it is representative of the shallowest depth reached by the hypoxic waters (Figure 3)." to "The 275 m isobath is chosen because it is reached each year by the hypoxic waters (Figure 3). This isobath

represents a more conservative estimate than using 250 m, and leads to an underestimation of the hypoxic zone in some years.".

- Finally, over the years, we have carried out several transects perpendicular to the main axis of the Laurentian Channel. We have looked at these as well as at additional transects from the BioChem dataset, and they consistently show that the layer of hypoxic waters extends, nearly at the same depth or on the same isopycnal, throughout the width of the Laurentian Channel, even where the estuary widens into the gulf, including at the most seaward station where hypoxic waters were detected in 2021. These cross-channel transects do not provide a very high resolution of the shape of the hypoxic zone close to the edges of the channel, as they only included 3 to 5 stations over the width of the channel. Yet, the data reveal that, as mentioned above, the shape of the hypoxic zone is not markedly domed.

We will to add a short discussion of these limitations in the Method section of the paper, that would read as follows:

"The real shape of the oxycline is likely domed close to the edges the Laurentian Channel, where it intersects the seafloor, under the action of turbulent mixing and benthic respiration. This doming is, however, limited to the vicinity of the seafloor, as suggested by transects perpendicular to the Channel, and the resulting error is expected to be small."

Notwithstanding, we thought about how we could improve the estimate.
First, as suggested by the reviewer, we could derive the depth of the oxycline along the channel and use that depth. We verified whether the depth of the top of the hypoxic layer varied systematically with distance along the Laurentian Channel (the hypoxic layer extending, to shallower depths closer to the head of the channel compared to within the gulf). We found no clear trend.
Second, since the sampling resolution is relatively coarse in the estuary and gulf, we could use a modelled, high-resolution climatology to look at the shape of the isopycnals. We are however not aware of the existance of such a biogeochemical model that could reproduce the shape of the oxycline. As mentioned above, based on existing perpendicular (cross-channel) transects,we suspect that this approach will not significantly improve the accuracy of our estimate.
Note that we do not claim to determine the volume of hypoxic waters but the two-dimensional shadow that the hypoxic waters cast along the Laurentian Channel, irrespective of the depth at which the hypoxic waters are found. Figure 5 clearly illustrates that the tongue of hypoxic waters is typically centered between 250 and 300 m but hypoxic waters can be found at shallower depths and does not extend to the bottom as it spreads seaward.

**RC1: Apparently, most of the profiles are from spring and summer, but there appears to be no filter on which profiles are actually used or that seasonal changes in oxygen concentrations are taken into account. Were all data within a year (mainly spring and summer) pooled disregarding seasonal differences? The seasonal variability adds further to the uncertainty and potentially adds bias to the estimates.**

The reviewer is right, most of the surveys were conducted in spring and summer, winter surveys were only conducted on three consecutive years in recent years (2018, 2019, 2020), and the data were aggregated per year. Nevertheless, beyond the mean annual variations, no distinct seasonal variations could be discerned. Although one might expect to find slightly lower bottomwater oxygen concentrations in late summer or fall, after the spring bloom autochthonous organic matter settles to the seafloor, the available data show no consistent seasonality (see figure below in which each coloured star represents a given year). This is not unexpected because the deep, hypoxic waters, are isolated from the atmosphere by the Cold Intermediate Layer.

We will explicitly indicate that the data are aggregated per year, and justify this approach by noting that available data show no clear seasonality and that this study focuses on yearly variations. To further justify our assumption, we will substitute Fig. 3 for a version in which the data are aggregated monthly instead of yearly.

[Figure]

**RC1: Finally, there seems to be cross-sectional variation in oxygen concentrations across the channel (cf. Fig. 2), so how was this taken into account when interpolating spatially along the axis of the channel? Was there always a clear spatial oxygen gradient or could there be cases, where hypoxia was observed beyond (further out) stations without hypoxia, i.e. did a more irregular oxygen pattern ever occur? Overall, there is a general lack of clarity in the description of the data processing.**

Figure 2 shows lowest dissolved oxygen concentrations at each sampled station, along the Laurentian Channel, between 1995 and 2021. We aggregated the data for individual years, irrespective of the sampling times, to focus on the yearly trends. We show the lowest sampled oxygen concentration, for any station that samples deeper than 200 m. Hence, at some stations, the measurement shown might have been sampled above the hypoxic layer. The presence of some non-hypoxic concentrations (blue dots) between hypoxic measurements (pink dots) is a artefact of the aggregation of data from different cruises, i.e., waters sampled at different times of the year and at different depths, and of the fact that some stations close to the Laurentian Channel's edges are shallower than the hypoxic zone.

First, we will modify Figure 2 to address this comment. We will use different symbols to identify samples shallower than 250 m, and thus highlight that the non-hypoxic water are found at shallower depths. For example, in the 2009 map shown below, most of the blue stations in the estuary are diamonds, which show samples between 200 and 250m, while circles show samples below 250 m. To better explain the data aggregation and to include this new

information, we will modify the caption of Fig. 2 to write the following:
"Maps of all oxygen samples collected in each year, since 1995, for every station sampled deeper than 200 m. The color indicates the lowest oxygen concentration sampled over the water column. The symbol indicates if the deepest sample is located between 200 and 250 m (diamond) or below 250 m (circle). Hypoxic waters are represented in magenta. The thin black lines delineate the 275 m isobath."

[Figure]

Second, we will rewrite the text to clearly explain that our estimate is for the observed **maximal hypoxic area reached in any given year**. It would not be possible to provide an average area for a given year, as the water column was not sampled equally every season.

**RC1 : Second, I do not think the authors provide a compelling case when arguing for changes in the inflowing Atlantic water masses affecting with different temperature and oxygen properties. Since this is a focal point for the manuscript, I strongly suggest (actually, a requirement) that more information and support for this is provided in the manuscript (and not just referencing some of their own previous work). What is causing the increased inflow of NACW (changes in AMO or another climate index)? What are the specific temperature and oxygen properties for the different water masses that normally ventilate the bottom layer? Are these changes visible at the outermost stations in the St. Lawrence channel? Can the changing water masses at the mouth be traced towards the head of the LSLE (it should be possible as the residence time is stated to be 4-7 years)? With such a long residence time for the gulf and estuary, that would lead to some mixing of the different water masses, how come there is such an abrupt change bw 2015 and 2016? The authors need to substantiate this conclusion much better. I am left with more questions than convictions from reading the manuscript.**

Most if not all these questions are answered in detail in Jutras et al. (JGR-Oceans, 2020). We feel that it would be unreasonable to repeat all this information here as it would unduly lengthen the manuscript. The main objective of the present paper is to present the temporal variations in spatial extent of the hypoxic zone. The other objective is to highlight the sudden drop in oxygen since publication of the Jutras et al. (2020), in which the most recent data are from 2018. Moreover, the mechanism that explains the 2018-2021 oxygen decline is the same as the one that Jutras et al, (2020) identified for the 2008-2018 decline. Hence, no new information would come from further discussion on these causes. Alternatively, we will provide a more detailed summary of the method (i.e., e-OMP), the properties of the source/parental waters to the bottom

waters of the Laurentian Channel, how their relative contributions have varied over the last few decades and why, as well as the relative contribution of in-situ (pelagic or benthic) respiration as the bottom waters flow inland from the mouth to the head of the Laurentian Channel.
* * *
**Detailed comments:**

**RC1: L. 30-32: The assertion that hypoxia and anoxia occur naturally in the mentioned systems is not entirely correct. To my knowledge, Chesapeake Bay did not experience hypoxia before the arrival of Europeans and the following deforestation. The Baltic Sea has had periods with hypoxia/anoxia in the geological past, but the spatial extent was never at the magnitude of the current spread. This sentence should be modified to avoid potential misinterpretations that the current spread of hypoxia is natural, which it is not!**

The sentence will be modified to avoid potential misinterpretations. We propose the following substitution: "Hypoxia and anoxia occur naturally in many coastal environments with restricted circulation, such as fjords and embayments, but hypoxia in more open coastal and estuarine areas appears to be on the rise due to anthropogenic nutrient loading and coastal eutrophication (e.g. Saanich Inlet in British Columbia, Bedford Basin in Nova Scotia, Chesapeake Bay in Maryland, shelf region of the northern Gulf of Mexico, the Kattegat in the Baltic Sea, the Bengali Current in western Africa, and the coastal area of the Changjian River/Estuary in the East China Sea; Bindoff et al., 2019; Breitburg et al., 2018; Gilbert et al., 2010; Rabalais et al., 2010; Li et al., 2002)."

**RC1: L. 92: 'distinct' should be 'discrete'.**

A much better choice of word. It will be substituted in the revised manuscript.

**RC1: L. 128-130, Figure 4: Were these observations (for linear regression) made at the same depth? If not, then the regression and the results from it do not make sense. It is commonly seen in such monitoring that sampling depths are getting closer to the bottom where oxygen gradients can be strong in more recent years with the development of more advanced CTD's. The authors need to analyse if depths are the same throughout the time series, and if the seasonal sampling time is more or less the same. It is also important to assess whether the samples represent the same salinity to ensure that the waters have the same properties. Why have the authors decided to show only data from the head of the LSLE? If the hypothesis of lower oxygen in the inflowing Atlantic water is correct then the same pattern should be largely paralleled throughout the channel at specific locations. This would increase the support for the hypothesis.**

Figure 4 reports the minimum dissolved oxygen concentrations (DO) observed near the head of the LSLE. Until the early part of 2000, and apart from data near Cabot Strait, no other historical data were available for the deep waters of the Laurentian Channel. The early measurements (1935-1980) in the LSLE are scarce and were made only in summer at depths exceeding 250 m but not necessarily at the same depth. Hence, we agree with the reviewer that the regression is a very crude estimate of the average rate of DO decline for the period 1935-1980. We will

caution the reader about the reliability of the regression in the revised manuscript by noting: "This historical reconstruction offers a crude estimate of the trend in deep oxygen concentration, given that the exact sampling depth, although always below 250m, might have varied over the decades". We will also add the following at the end of the method section: "Finally, we use oxygen concentration measured in the 1930s by clerics from Université Laval to extend the time series (Figure 4)."

For as long as we have a historical record, the salinity and temperature of the bottom waters of the Laurentian Channel have increased steadily as the relative contribution of the parental waters (Labrador Current and North-central Atlantic Water) have evolved in time. We do not discuss these variations here because they were previously described in Gilbert et al. (2005) and in Jutras et al. (2020b). In these papers the authors show that the temperature and salinity do not directly track the DO concentrations because eutrophication and microbial respiration contribute to the deoxygenation. We will add this information at L158. Transects of the two parental water masses along the Laurentian Channel are shown in Jutras et al. (2020b). It is, however, interesting to note that, despite changes in bottom-water temperatures and salinities, the depth of the oxygen minimum has not changed considerably over the last 90 years and sits at approximately 250 m depth or the 27.25 (kg m$^{-3}$) isopycnal. Finally, we focus on the head of the LSLE because the lowest DO concentrations are found in this region where the strongest consequences on this marine ecosystem are likely to be observed.

**RC1: L. 154-158: The argumentation here is essential for understanding the changes in oxygen in the LSLE, but instead of showing any evidence the authors refer to their previous study and one from 2005, neither of those contain the more recent data that motivated the study according to the introduction. It is necessary for the authors to present updated datasets for these patterns of mixing on the shelf. Without such data this argumentation remains unconvincing.**

As noted above, we feel that it would be unreasonable to repeat all this information here as it would unduly lengthen the manuscript and because the goal of the present manuscript is not to discuss in detail the deoxygenation mechanisms but rather to highlight the sudden decline and estimate the change in the areal extent of the hypoxic zone.. Alternatively, we will provide a more detailed summary of the method (i.e., e-OMP), the properties of the source/parental waters to the bottom waters of the Laurentian Channel, how their relative contributions have varied over the last few decades and why, as well as the relative contribution of in-situ (pelagic or benthic) respiration as the bottom waters flow inland from the mouth to the head of the Laurentian Channel. The most recent field data are presented in Figure 6 and results of the e-OMP analysis of these data are presented in Figure 7.

**RC1: Figure 6: What are the observations showing, i.e. are they observed measurements or means from several cruises or ....?**

Results presented in Figure 6 are a compilation of overlapping field measurements conducted in 2021: 1) from the head of the LSLE to the entrance to the Gulf of St. Lawrence near Pointe-des-Monts from August 25 to 30, 2021 and 2) from Rimouski and throughout the Laurentian, Anticosti and Esquiman Channels, all the way to Cabot Strait from October 23 to 29, 2021. This information will be added to the figure caption, and different symbols will be used to identify the two different surveys.

**RC1: Figure 7: The authors cannot present important results without providing more explicit information on how these were computed. It is insufficient to reference Jutras et al. 2020b, assuming that the approach in that study is well known.**

As noted above, we feel that it would be unreasonable to repeat all this information here as it would unduly lengthen the manuscript and because the goal of the present manuscript is not to discuss in detail the deoxygenation mechanisms but rather to highlight the sudden decline and estimate the change in the extent of the hypoxic zone. Alternatively, we will provide a more detailed summary of the method (i.e., e-OMP), the properties of the source/parental waters to the bottom waters of the Laurentian Channel, how their relative contributions have varied over the last few decades and why, as well as the relative contribution of in-situ (pelagic or benthic) respiration as the bottom waters flow inland from the mouth to the head of the Laurentian Channel.

**RC1: L. 175: Should be 'NACW'.**

Thanks for picking this up, it will be corrected in the revised manuscript.

**RC1: L. 187: 'temperature'**

Thanks for picking this up, it will be corrected in the revised manuscript.

Finally, we noticed that the data above 150 µmol/kg are presented in magenta instead of yellow on Fig. 2. This error will be corrected in the revised manuscript.

---

## Author Comment (AC2)

**Response to reviewer #2 on manuscript egusphere-2022-1090 titled "Temporal and spatial evolution of bottom-water hypoxia in the Estuary and Gulf of St. Lawrence". Comments posted Dec 9, 2022.**

Note that many of our responses to RC2 are derived from our answers to RC1 since very similar comments/recommendations were made by both reviewers.

**RC2:** I found this to be a clearly written paper that efficiently reports a new piece of science around hypoxia in the system studied. My comments are primarily made to encourage the authors to add more detail to some of the methods and many of the figure captions, so some of the key details are clearly communicated.

We thank the reviewer for his(her) supportive and helpful comments.

**RC2:** Figure 2: Are these concentrations for all samples collected, or from a consistent timeframe? It is worth specifying that here.

We agree with the reviewer that Figure 2 and its caption were confusing. Figure 2 shows the lowest dissolved oxygen concentrations at all the stations sampled in a given year, possibly from different surveys, along the Laurentian Channel, between 1995 and 2021. We aggregated the data for individual years, irrespective of the sampling times, to focus on the yearly trends. Most of the stations were sampled in the summer, with winter surveys in recent years (2018, 2019, 2020). We will modify the manuscript to explicitly indicate that the data are aggregated per year and that the maps show data from different surveys. We will also substitute Fig. 3 for a version that aggregates the data per season instead of yearly (see below). We could also include a new inset showing a histogram of the number of stations sampled every month from 1995 to 2021.

[Figure]

New version of Fig. 3, where the data are aggregated per season.

In addition to the changes related to sampling time, we modify Figure 2 to better represent the sampling depths. The figure shows the lowest sampled dissolved oxygen (DO) concentration, for any station for samples deeper than 200 m. Hence, at some stations, the measurement shown might have been sampled above the hypoxic layer. To avoid further confusion, we will use different symbols to identify samples shallower than 250 m, and thus highlight that the non-hypoxic waters (DO > 62.5 µM) are found at shallower depths. For example, in the 2009 map shown below, most of the blue stations in the estuary are diamonds, which identify samples recovered between 200 and 250m, while circles show samples collected below 250 m.

To better explain the data aggregation and to include this new information, we will modify the Fig. 2 caption to the following:
"Maps of all dissolved oxygen concentrations for samples collected in each year, since 1995, for every station deeper than 200 m. The color indicates the lowest oxygen concentration sampled over the water column. The symbol indicates if the deepest sample is located between 200 and 250 m (diamond) or below 250 m (circle). Hypoxic waters are represented in magenta. The thin black lines delineate the 275 m isobath."

[Figure]

RC2: Line 132: Perhaps replace "alarmingly fast" with "rapidly"? The current language reads a little awkwardly and is somewhat sensational.

We will tame down the vocabulary and substitute "alarmingly fast" by "rapidly" in the revised manuscript. Irrespective, after more than 20 years of near status quo, the sudden decrease in minimum dissolved oxygen (DO) concentrations is alarming. According to more recent measurements (September 2022), minimum DO may have dropped further to 27-30 µM.

RC2: Line 134: Is strikes me that you could argue that temperature rose rapidly in the past two years (as oxygen declined rapidly), but you really only speak to the progressive longer term trend here. This comment is also relevant in the conclusions on line 216.

This is a good point. As the minimum DO concentrations decreased rapidly, the water temperature also increased quickly, from 5.6 to 6.2ºC in the same parcel of water, between 2018 and 2021, reaching in 2020 nearly 7ºC in the western Gulf of St.

Lawrence. We will add this information in the revised manuscript at line 134: "bottom-water temperatures in the LSLE and the GSL have increased progressively from ~3°C in the 1930's to nearly 7°C in 2020, **thus showing a rapid 1°C increase from 2019 to 2020**."
At L216, we will change "bottom-water temperatures have increased steadily" for "bottom-water temperatures increased by one degree in only one year".

**RC2:** Figure 4: What does the grey area in top panel represent?

The grey band is a visual guide and crude estimate of the temporal trend of the minimum bottom-water oxygen concentrations in the Lower St. Lawrence Estuary over the period of available field measurements.

**RC2:** Figure 5: Can you specify when these samples were taken each year? Is it possible that they fluctuate enough over the season that a snapshot might miss some intra-seasonal variations? If so, this would be a relevant discussion point to communicate some of the uncertainty in the study.

Like Figure 2, this figure shows an aggregation of all samples taken in any given year. We will modify the caption to indicate the periods (days and months) over which the data presented in Figure 5 were acquired in the revised caption. Most of the surveys were conducted in spring and summer, winter surveys were only conducted on three consecutive years in recent years (2018, 2019, 2020), and the data were aggregated per year in Figs 2, 3 and 5. Nevertheless, beyond the mean annual variations, no distinct seasonal variations could be discerned. This is not unexpected because the deep, hypoxic waters, are isolated from the atmosphere by the Cold Intermediate Layer. Only the spring bloom of autochthonous organic matter could affect deep oxygen levels on a seasonal scale, but the available data show no consistent seasonality (see figure below in which each coloured star represents a given year).
We will modify the introduction to include this information. We will modify L69-70 from "During most of the ice-free season, the water column of the LSLE can be described as a three-layer system on the basis of its thermal stratification." to "The LSLE is a strongly stratified system that can be described as a three-layer system.", and we will add the following at L74: "bottom layer (>150 m deep) flows sluggishly landward, **isolated from the atmosphere** (Dickie & Trites, 1983), for 4 to 7 years from the continental shelf-break to the head of the Laurentian Channel (Bugden et al., 1991; Gilbert, 2004). "
As mentioned above, we will also explicitly indicate that the data are aggregated per year, and justify this approach by noting that available data show no clear seasonality and that this study focuses on yearly variations. To further justify our assumption, we will substitute Fig. 3 for a version in which the data are aggregated monthly instead of yearly.

[Figure]

**RC2:** Figure 6: Is there an explanation why there is 20-50% LCW fraction in the ~275 m water between 200 and 500 km? The only reference to this figure is that LCW is now almost null, but this seemingly residual or preserved LCW fraction was not discussed, and I think it should be.

Since the bottom waters require 4-7 years to transit from the mouth of the Laurentian Channel to its head (at the head of the LSLE), the layer of 20-50% LCW that appears at ~250m depth at the landward part of the transect could be bottom water that is several years older and richer in LCW than waters now entering the Gulf at Cabot Strait,.The null LCW fraction seen above and below the LCW-enriched layer is expected, since below we find the NACW, and above, the CIL. The presence of the LCW, distinct for the NACW, was first resolved in Jutras et al. (2020b) upon a more refined e-OMP analysis of available data. The analysis revealed that the LCW and NACW that make up the bottom waters of the Laurentian Channel are not perfectly mixed and a layer of water richer in LCW sits on top of a NACW-rich layer in the Lower St. Lawrence Estuary (LSLE). We agree that a discussion of the uncertainty of this method is missing, and we will add a few sentences on this subject in the text. We will also add an explanation on the observed 20-50% LCW fraction either in the caption or the main text of the manuscript.

**RC2:** Line 160: Are there data to report here about organic matter and nutrient exports? Minimally it would seem relevant to report a quantitative change, or maximally a figure of the long-term data.

A limited historical record (1995-2012) of organic matter and nutrient exports from the St. Lawrence River to the Upper St. Lawrence Estuary was published by Hudon et al. (1997), but no clear trend could be identified over this period.  Unfortunately, we could not find a multi-decadal time series of nutrient loading for the St. Lawrence River, as is available for the Mississippi River (Turner et al., 2007). The Mississippi River record reveals a large anthropogenic increase in nutrient loading since about 1970. Given the geographic proximity of the two drainage basins, and the similar agricultural practices and industrial activities, it would be reasonable to assume that nutrient loading to the St.

Lawrence River has also increased over the last decades. Indirect accounts of nutrient loading certainly support this hypothesis, like the increase of nitrogen and phosphate fertilizer sales in Quebec and Ontario (Statistics Canada, 2018), as well as the, respectively 3.8 to 4.5-fold, increase in the estimated net anthropogenic P and N inputs to watersheds of the St. Lawrence Basin over the last century (Goyette et al., 2016). This impact of nutrient loading to the Lower St. Lawrence Estuary (LSLE) was addressed in detail in Jutras et al. (2020a) in which the authors present a three-layer box model and budget of nutrients in the LSLE. In this paper, they conclude that the majority of nutrients delivered to the LSLE originate from upwelling of nutrient-rich waters at the head of the channel. Whereas bottom-water nutrient concentrations have increased noticeably since at least the early 1960's, they reached near steady-state values in the 1990's. We will detail these findings in the revised manuscript instead of simply citing these references at the end of L161.

**RC2:** Line 185-190: I appreciate that the methodology has been presented before (Jutras et al. 2020b) to partition the delta DO among driving factors, but I think a brief explanation of the approaches here would help make this paper stand alone.

We feel that it would be unreasonable to repeat details of this methodology here as it would unduly lengthen the manuscript. The main objective of the present paper is to present the temporal variations in spatial extent of the hypoxic zone. The other objective is to highlight the sudden drop in dissolved oxygen (DO) concentrations since publication of the Jutras et al. (2020), in which the most recent data are from 2018. Identification of the mechanism responsible for this sudden drop in DO does not represent a new scientific contribution, since it is the same as for the 2008-2018 period covered in Jutras et al, (2020). Hence, no new information would come from further discussion on these causes. Alternatively, we will provide a more detailed summary of the method (i.e., e-OMP), the properties of the source/parental waters to the bottom waters of the Laurentian Channel, how their relative contributions have varied over the last few decades and why, as well as the relative contribution of in-situ (pelagic or benthic) respiration as the bottom waters flow inland from the mouth to the head of the Laurentian Channel.

**RC2:** Figure 8: can you express what error bars represent here? How were they calculated?

The error bars represent the uncertainty associated with the calculation of the areal extent of the hypoxic zone, as described at L110-111. The uncertainty is, in great part, a consequence of the low spatial resolution sampling, and more specifically the extrapolation of the area between the last hypoxic station and the first non-hypoxic station along the Laurentian Channel. We will add a reference to L110-111 in the revised caption.

---

## Author Response (AR1)

**Response to reviewers on manuscript egusphere-2022-1090 titled "Temporal and spatial evolution of bottom-water hypoxia in the Estuary and Gulf of St. Lawrence"**

**Jan 6, 2023**
* * *
**Reviewer #1**

**RC1: This manuscript evaluates the temporal changes in hypoxia in the St. Lawrence River Estuary and Gulf from the earliest measurements in the 1930s to present. This manuscript is apparently an update of a previous paper by the main author (Jutras et al. 2020, JGR) with the main conclusion that hypoxia expanded drastically in recent years reaching almost 10,000 km2 in 2021. This conclusion may not be wrong but the authors have not managed to provide compelling evidence to support it!**

We thank the reviewer for his(her) incisive and helpful comments.

**RC1: First, the method used for assessing the extent of hypoxia is very coarse. The authors determine the area deeper than 275 m from the mouth of the St. Lawrence Channel to the farthest station with hypoxia. This approach does not consider that oxygen profiles are irregularly distributed along the gradient, which in combination with the gradual broadening of the channel can give quite variable results. The authors do acknowledge this limitation, but instead they should try to improve the spatial integration of the profiles by developing a more sound data processing approach, e.g. by fitting the oxycline as function of depth along the channel gradient. Moreover, hypoxic conditions can be observed at shallower depths than 275 m (cf. Fig. 3) and it can also be deeper, so why make this simplistic approach of considering the area deeper than 275 m. How do the authors define hypoxic conditions when oxygen concentrations are changing with depth? Importantly, more precise estimates can be obtained by minor improvements in the data processing through formulating an appropriate model.**

We agree with the reviewer that the method used to estimate the areal extent of hypoxia is rather coarse. Below, we explain why using the 275 m isobath is currently the most precise way to estimate the area of the hypoxic zone.

- The real shape of the hypoxic zone does not follow exactly the 275 m isobath, especially close to the edges of the Laurentian Channel, where turbulent mixing and benthic activity might dome the oxybaths. This possible vertical shift in the oxybaths, where the waters intercept the seafloor, is one of the uncertainties in our method. However, the current spatial sampling in the St. Lawrence Estuary is too sparse to assess these effects. The system is typically sampled in the middle of the Laurentian Channel, at about a dozen stations along the channel from its mouth to its head. Transects perpendicular to the channel are sometimes carried out, but for a few stations only, every ~10 km or so. We have looked at these, and they consistently show that, on this spatial scale, the layer of hypoxic waters extends nearly at the same depth or on the same isopycnal throughout the width of the Laurentian Channel, even

where the estuary widens into the gulf, including at the most seaward station where hypoxic waters were detected in 2021. These cross-channel transects, such as the one shown below, do not provide a very high resolution of the shape of the hypoxic zone close to the edges of the channel, but they reveal that the shape of the hypoxic zone is not markedly or consistently domed. Moreover, mixing is expected to dome the hypoxic zone upward, while benthic oxygen consumption is expected to dome the hypoxic zone downward. Consequently, the two effects may partially compensate each other, and we do not expect the overestimation or underestimation to be very significant.

At the western limit of the hypoxic zone, at the head of the Laurentian Channel, we also expect mixing to carry oxygen deeper in the water column, leading to a slight overestimation of the hypoxic area when using the 275 m isobath. However, the head of the channel represents a small fraction of the hypoxic region.

- Another point is that, whereas the hypoxic layer usually reaches up to a depth of 250 m (see Figures 3 and 5), we use the 275 m isobath, a more conservative choice that might lead to an underestimation of the hypoxic zone.

[Figure]

*Oxygen concentrations along a transect perpendicular to the Laurentian Channel, within the Lower St. Lawrence Estuary*

We added a short discussion of these limitations in the Method section of the revised manuscript, at L122-L130, that reads as follows:

"*This spatial extent calculation method has a number of limitations. First, even if the hypoxic waters often reach up to 250 m depth, we use a conservative value of 275 m for the isobath, as it is reached every year (Figure 3). This choice leads to an underestimation of the spatial extent of the hypoxic zone in some years. Second, the real shape of the oxycline is likely domed close to the edges the Laurentian Channel, where it intersects the seafloor. There, turbulent mixing will dome the oxycline downward, while benthic respiration will dome it upward. The exact shape of the oxycline is not known, due to the weak spatial sampling rate in the St. Lawrence estuarine system. Yet, low resolution (every ~10 km) transects perpendicular to the Laurentian Channel show a flat oxycline, suggesting that the doming is limited to the near edges of the channel. Hence, the error on estimates of the spatial extent of oxygen from an isobath will be small.*"

Notwithstanding, we thought about how we could improve the estimate.

First, as suggested by the reviewer, we could derive the depth of the oxycline along the channel and use that depth. We verified whether the depth of the top of the hypoxic layer varied systematically with distance along the Laurentian Channel (the hypoxic layer might extend to shallower depths closer to the head of the channel compared to within the gulf). We found no clear trend.

Second, since the sampling resolution is relatively coarse in the estuary and gulf, we could use a modelled, high-resolution climatology to look at the shape of the oxycline. We are, however, unaware of the existence of a biogeochemical model that could provide such an output. As mentioned above, based on existing perpendicular (cross-channel) transects, we suspect that this approach will not significantly improve the accuracy of our estimate.

Note that we do not claim to determine the volume of hypoxic waters, but the two-dimensional shadow that the hypoxic waters cast along the Laurentian Channel, irrespective of the depth at which the hypoxic waters are found. Figure 5 clearly illustrates that the tongue of hypoxic waters is typically centered between 250 and 300 m depth but that hypoxic waters can be found at shallower depths. Furthermore, the tongue of hypoxic waters does not extend to the bottom as it spreads seaward.

**RC1: Apparently, most of the profiles are from spring and summer, but there appears to be no filter on which profiles are actually used or that seasonal changes in oxygen concentrations are taken into account. Were all data within a year (mainly spring and summer) pooled disregarding seasonal differences? The seasonal variability adds further to the uncertainty and potentially adds bias to the estimates.**

The reviewer is right, most of the surveys were conducted in spring and summer, winter surveys were only conducted on three consecutive recent years (2018, 2019, 2020), and the data were aggregated per year. Beyond the mean annual variations, no distinct seasonal variations could be discerned. This is not unexpected because the deep, hypoxic waters, are isolated from the atmosphere by the Cold Intermediate Layer. One might expect to find slightly lower bottom-water oxygen concentrations in late summer or fall, after the spring bloom autochthonous organic matter settles to the seafloor, but the available data show no consistent seasonality (see figure below, in which each colored star represents one year).

[Figure]

- In the revised manuscript, we first specify that most of the profiles were taken in the spring, summer and fall by adding the text in bold in the Method section, at L81:
  "*The first data set includes measurements we acquired **mostly during** the spring and summer between 2003 and 2021 **and in the winder from 2018 to 2020** [...]*"
  and at L94
  "*The second data set was extracted from the Bio-Chem database [...] and covers the Gulf and St. Lawrence Estuary from 1967 to 1972, and from 1991 to 2018, **from spring to fall***".
- Then, we more explicitly indicate that the data was aggregated per year, by adding the following sentence in the Method section at L101:
  "*We combine these three data sets, and aggregate the data per year, to focus on the inter-annual variability.*"
  and later at L116
  "*We aggregate all the measurements made during each year, and hence the estimates represent the maximal hypoxic area reached each year, during the sampled periods. Based on the available data, the spatial extent does not appear to vary seasonally.*",
  and by modifying the caption of Fig. 2 so it reads as follows:
  "*Maps showing a compilation of all dissolved oxygen (DO) samples collected every year since 1995, for every station sampled deeper than 200 m. A map may contain data from multiple surveys carried out throughout that year. [...]*".
- We justify this by noting that available data show no clear seasonality and that this study focuses on yearly variations, by adding the following at L102:
  "*The seasonality in air temperature and river runoff do not affect the bottom-water properties of the Laurentian Channel, as they are isolated from the surface by the intermediate layer (CIL). Only the spring bloom and delivery of autochthonous organic matter could affect the bottom-water oxygen levels on a seasonal scale, but available data show no consistent seasonality.*".
- To further justify our assumption, we substituted Fig. 3 for a version in which the data are aggregated monthly instead of yearly.

**RC1: Finally, there seems to be cross-sectional variation in oxygen concentrations across the channel (cf. Fig. 2), so how was this taken into account when interpolating spatially along the axis of the channel? Was there always a clear spatial oxygen gradient or could there be cases, where hypoxia was observed beyond (further out) stations without hypoxia, i.e. did a more irregular oxygen pattern ever occur? Overall, there is a general lack of clarity in the description of the data processing.**

Figure 2 shows the lowest dissolved oxygen concentrations at each sampled station, along the Laurentian Channel, between 1995 and 2021. We aggregated the data for individual years, irrespective of the sampling times, to focus on the yearly trends. We show the lowest sampled oxygen concentration for all stations from which samples were recovered deeper than 200 m. Hence, at some stations, the measurement shown might have been sampled above the hypoxic layer, especially near the coasts where the water is shallower than the hypoxic zone. About the presence of some non-hypoxic concentrations (blue dots) between hypoxic measurements (pink dots), it is an artefact of the aggregation of data from different cruises, i.e., waters sampled at different times of the year and at different depths.

To address this, we first modified Figure 2 and use different symbols to identify samples shallower than 250 m (diamonds) and deeper than 250 m (circles). This highlights that the non-hypoxic waters are actually found in shallow areas. For example, in the 2009 map shown below, most of the blue stations in the estuary are diamonds. We added the following to the caption of Fig. 2:
"*The color indicates the lowest oxygen concentration sampled over the water column, with hypoxic waters in magenta. The symbols indicate if the deepest sample is located between 200 and 250 m (diamond) or below 250 m (circle).*"

[Figure]

Second, we specified in the text that, because of the aggregation of data from different seasons, our estimate is for the observed maximal hypoxic area reached in any given year, at L116:
"*We aggregate all the measurements made during each year, and hence the estimates represent the maximal hypoxic area reached each year, during the sampled periods.*"
It would not be possible to provide an average area for a given year, as the water column was not sampled equally every season.

**RC1 : Second, I do not think the authors provide a compelling case when arguing for changes in the inflowing Atlantic water masses affecting with different temperature and oxygen properties. Since this is a focal point for the manuscript, I strongly suggest (actually, a requirement) that more information and support for this is provided in the manuscript (and not just referencing some of their own previous work). What is causing the increased inflow of NACW (changes in AMO or another climate index)? What are the specific temperature and oxygen properties for the different water masses that normally ventilate the bottom layer? Are these changes visible at the**

**outermost stations in the St. Lawrence channel? Can the changing water masses at the mouth be traced towards the head of the LSLE (it should be possible as the residence time is stated to be 4-7 years)? With such a long residence time for the gulf and estuary, that would lead to some mixing of the different water masses, how come there is such an abrupt change bw 2015 and 2016? The authors need to substantiate this conclusion much better. I am left with more questions than convictions from reading the manuscript.**

Most if not all these questions are answered in detail in Jutras et al. (JGR-Oceans, 2020). We feel that it would be unreasonable to repeat all this information here as it would unduly lengthen the manuscript. The main objective of the present paper is to present the temporal variations in the spatial extent of the hypoxic zone. The other objective is to highlight the sudden drop in the minimum dissolved oxygen concentration since publication of the Jutras et al. (2020), in which the most recent data are from 2018. Moreover, the mechanism that explains the 2018-2021 oxygen decline is the same as the one that Jutras et al. (2020) identified for the 2008-2018 decline. Hence, no new information would come from further discussion of these causes. To make the objective of the paper clearer, we reorganized the result section of the paper. First, we moved section 3.4 (Areal extent of the hypoxic zone) to section 3.2. Second, we combined sections 3.2 'Causes' and section 3.3 'Sources of water' into a single section, removing some repetitions and emphasizing what the original contributions of this manuscript are by adding, at L190:
"*The causes of deoxygenation in the Lower St. Lawrence Estuary from the 1930s to 2018 were discussed in detail elsewhere (e.g. Jutras et al., 2020b; Gilbert et al., 2005) and, thus, we here summarize their contribution to the most recent decline of minimum dissolved oxygen (DO) concentrations (2018 to 2021).*"
and at L136:
"*Here, we apply the eOMP method used in Jutras et al. (2020b) (Section 2) to reconstruct the causes of the 2019 to 2021 oxygen decline (Figure 8).*"

To specifically answer the reviewer's questions, we made the following modifications:

- What is causing the increase in the NACW is an on-going topic of research in the community, and we added additional details on the current understanding of the phenomenon at L157: "*The recent increase in the amount of Gulf Stream water in the Slope Sea and feeding the LC is believed to be due to a slow-down of the Atlantic Meridional Overturning Circulation (AMOC, New et al., 2020), as well as to an increased retroflection of the Labrador Current towards the east in the vicinity of the Grand Banks in response to a stronger Labrador Current (Jutras et al., 2020b) and strong westerly winds over the Labrador Shelf (Holliday et al., 2020), both of which are related to a large-scale adjustment of the circulation patterns in the western North Atlantic (Jutras et al., in revision).*"
- We give more details about the parent water mass properties at L209:
"*The former are colder and carry more DO than the latter (-0.7°C to 3.2°C and 310 to 280 µmol/kg of DO for the LCW, compared to 4.4°C to 17.7°C and 155 to 250 µmol/kg of DO for the NACW. Jutras et al., 2020b)*"
- For more details about how the water masses mix, we refer the reader more explicitly to Jutras et al. (2020b) at L236, since this topic is addressed in detail in that paper:
"*Here, we apply the eOMP method used in Jutras et al. (2020b) (Section 2) to reconstruct the causes of the 2019 to 2021 oxygen decline (Figure 8). We refer the reader to the original paper for details on the properties of the different water masses, their distribution along the*

*Laurentian Channel, and the temporal variability in their relative contributions to the bottom waters of the Laurentian Channel.*"

- We provide more details on the eOMP method, in the Method section at L108, by adding the text in bold: "***In this method, a set of linear equations that describe the properties ($S_P$, $\delta^{18}O(H_2O)$, temperature, alkalinity, dissolved oxygen and nutrient concentrations) of a parcel of water is used to determine*** *the relative contributions of the different water types* ***that make up that parcel of water, given a definition of these water types in terms of the available water properties. Unlike the T-S diagram method, the eOMP method accounts for diapycnal mixing and provides estimates of biogeochemical changes that occurred between the water type formation and the measurement locations. Details of the application of this method to the current data set can be found in Jutras et al. (2020b).***"

and, in the Results section 3.4 at L249:

"*Another 30% of the 2018 to 2021 deoxygenation is attributed to the increased biological oxygen consumption rates during the transit of waters along the Laurentian Channel in response to the temperature increase (Figure 8),* ***estimated from the empirical temperature dependence of bacterial respiration rate $Q_{10}$ (Jutras et al., 2020b; Genovesi et al., 2011; Bailey & Ollis, 1986)****. The remaining 10% decrease is due to an* ***increased oxygen consumption in the North Atlantic****, also likely due to higher water temperatures,* ***as estimated from the results of the eOMP analysis at the entrance of the Laurentian Channel****.*"

**Detailed comments:**

**RC1: L. 30-32: The assertion that hypoxia and anoxia occur naturally in the mentioned systems is not entirely correct. To my knowledge, Chesapeake Bay did not experience hypoxia before the arrival of Europeans and the following deforestation. The Baltic Sea has had periods with hypoxia/anoxia in the geological past, but the spatial extent was never at the magnitude of the current spread. This sentence should be modified to avoid potential misinterpretations that the current spread of hypoxia is natural, which it is not!**

The sentence has been modified to avoid potential misinterpretations:
"*Hypoxia and anoxia occur naturally in many coastal environments with restricted circulation, such as fjords and embayments, but hypoxia in more open coastal and estuarine areas appears to be on the rise due to anthropogenic nutrient loading and coastal eutrophication (e.g. Saanich Inlet in British Columbia, Bedford Basin in Nova Scotia, Chesapeake Bay in Maryland, shelf region of the northern Gulf of Mexico, the Kattegat in the Baltic Sea, the Bengali Current in western Africa, and the coastal area of the Changjian River/Estuary in the East China Sea; Bindoff et al., 2019; Breitburg et al., 2018; Gilbert et al., 2010; Rabalais et al., 2010; Li et al., 2002).*"

**RC1: L. 92: 'distinct' should be 'discrete'.**

A much better choice of word. It was substituted in the revised manuscript.

**RC1: L. 128-130, Figure 4: Were these observations (for linear regression) made at the same depth? If not, then the regression and the results from it do not make sense. It is commonly seen in such monitoring that sampling depths are getting closer to the bottom where oxygen gradients can be strong in more recent years with the development of more advanced CTD's. The authors need to analyse if depths are the same throughout the time series, and if the seasonal sampling time is more or less the same. It is also important to assess whether the samples represent the same salinity to ensure that the waters have the same properties. Why have the authors decided to show only data from the head of the LSLE? If the hypothesis of lower oxygen in the inflowing Atlantic water is correct then the same pattern should be largely paralleled throughout the channel at specific locations. This would increase the support for the hypothesis.**

Figure 4 reports the minimum dissolved oxygen concentrations (DO) observed near the head of the LSLE. Until the early part of 2000, and apart from data near Cabot Strait, no other historical data were available for the deep waters of the Laurentian Channel. The early measurements (1935-1980) in the LSLE are scarce and were made only in summer at depths exceeding 250 m but not necessarily at the same depth. Hence, we agree with the reviewer that the regression is a very crude estimate of the average rate of DO decline for the period 1935-1980. We caution the reader about the reliability of the regression in the revised manuscript by noting, in the caption of Fig. 4:
"*This historical reconstruction offers a crude estimate of the trend in bottom-water minimum DO concentrations, given that the exact sampling depth, although always below 250m, might have varied over the decades.*"
We also added the following at the end of the Method section, at L100 to clarify where the early data originate from:
"*The third data set contains oxygen concentrations measured by Winkler titration by clerics from Université Laval in the 1930s, and is used to extend the time series (Figure 4).*"

We focus on the head of the LSLE because the lowest DO concentrations are found in this region where the strongest consequences on this marine ecosystem are likely to be observed. It is true that the oxygen decline is visible in other areas of the estuary. A time series of the oxygen concentration at the entrance of the Laurentian Channel is shown in Jutras et al. (2020b). For as long as we have a historical record, the salinity and temperature of the bottom waters of the Laurentian Channel have increased steadily as the relative contributions of the parental waters (Labrador Current and North-central Atlantic Water) have evolved in time. We do not discuss these variations here because they were previously described in Gilbert et al. (2005) and in Jutras et al. (2020b). In these papers, the authors show that the temperature and salinity do not directly track the DO concentrations because eutrophication and microbial respiration also contribute to the deoxygenation. We added the text in bold at L213:
"***Notwithstanding, the temporal variability of bottom-water DO concentration does not exactly track the temperature and salinity variability, because*** *eutrophication (Jutras et al., 2020b; Thibodeau et al., 2006) and an increase of the microbial respiration rates of settling organic matter in response to the increase in bottom-water temperatures (Genovesi et al., 2010) explains much of the remaining oxygen decline.*"
Transects of the two parental water masses along the Laurentian Channel are shown in Jutras et al. (2020b), for LCW in Figure 7, and for oxygen in Figure 5. It is, however, interesting to note that, despite changes in bottom-water temperatures and salinities, the

depth of the oxygen minimum has not changed considerably over the last 90 years and sits at approximately 250 m depth or the 27.25 (kg m$^{-3}$) isopycnal.

**RC1: L. 154-158: The argumentation here is essential for understanding the changes in oxygen in the LSLE, but instead of showing any evidence the authors refer to their previous study and one from 2005, neither of those contain the more recent data that motivated the study according to the introduction. It is necessary for the authors to present updated datasets for these patterns of mixing on the shelf. Without such data this argumentation remains unconvincing.**

As noted above, we feel that it would be unreasonable to repeat all this information here as it would unduly lengthen the manuscript and because the goal of the present manuscript is not to discuss in detail the deoxygenation mechanisms but rather to highlight the sudden decline and estimate the change in the areal extent of the hypoxic zone. We modified the manuscript to make the goal of the paper more explicit, and provided some additional information on the method and on the results. Please see the list of modifications applied to the manuscript in response to a similar comment above.

**RC1: Figure 6: What are the observations showing, i.e. are they observed measurements or means from several cruises or ....?**

Results presented in Figure 6 are from a survey conducted from Rimouski, throughout the Laurentian, Anticosti and Esquiman Channels and all the way to Cabot Strait, from October 23 to 29, 2021. This information was added to the figure caption (text in bold): "*Transect of the LCW fraction along the Laurentian Channel from a survey conducted **between October 23 and 29, 2021**, from the head of the Laurentian Channel (left) to Cabot Strait (right). **The slightly offset points depict stations located along transects perpendicular to the channel.**"*

**RC1: Figure 7: The authors cannot present important results without providing more explicit information on how these were computed. It is insufficient to reference Jutras et al. 2020b, assuming that the approach in that study is well known.**

As noted above, we feel that it would be unreasonable to repeat all this information here as it would unduly lengthen the manuscript and because the goal of the present manuscript is not to discuss in detail the deoxygenation mechanisms but rather to highlight the sudden decline and estimate the change in the areal extent of the hypoxic zone. We modified the manuscript to make our goal more explicit, and provided additional information about the method and results. Please see the list of modifications applied to the manuscript in response to a similar comment above.

**RC1: L. 175: Should be 'NACW'.**

Thanks for picking this up, it was corrected in the revised manuscript.

**RC1: L. 187: 'temperature'**

Thanks for picking this up, it was corrected in the revised manuscript.

**Finally, we noticed that the data above 150 µmol/kg are presented in magenta instead of yellow on Fig. 2. This error was corrected in the revised manuscript.**
* * *
**Reviewer #2**

Note that many of our responses to RC2 are derived from our answers to RC1 since very similar comments/recommendations were made by both reviewers.

**RC2: I found this to be a clearly written paper that efficiently reports a new piece of science around hypoxia in the system studied. My comments are primarily made to encourage the authors to add more detail to some of the methods and many of the figure captions, so some of the key details are clearly communicated.**

We thank the reviewer for his(her) supportive and helpful comments.

**RC2: Figure 2: Are these concentrations for all samples collected, or from a consistent timeframe? It is worth specifying that here.**

We agree with the reviewer that Figure 2 and its caption were confusing. Figure 2 shows the lowest dissolved oxygen concentrations at all the stations sampled in a given year, possibly from different surveys, along the Laurentian Channel, between 1995 and 2021. We aggregated the data for individual years, irrespective of the sampling times, to focus on the yearly trends. Most of the stations were sampled in the summer, with winter surveys in recent years (2018, 2019, 2020). We modified the manuscript to explicitly indicate that the data are aggregated per year and that the maps show data from different surveys.

- We now specify that most of the profiles are taken in the spring, summer and fall by adding the text in bold in the Method section at L83:
  "*The first data set includes measurements we acquired **mostly during** the spring and summer between 2003 and 2021 **and in the winder from 2018 to 2020** […]*"
  and at L95
  "*The second data set was extracted from the Bio-Chem database [...] and covers the Gulf and St. Lawrence Estuary from 1967 to 1972, and from 1991 to 2018, **from spring to fall***".
- Then, we more explicitly indicate that the data were aggregated per year, by adding the following sentence in the Method section, at L101:
  "*We combine these three data sets, and aggregate the data per year, to focus on the inter-annual variability.*"
  and later at L102:
  "*We aggregate all the measurements made during each year, and hence the estimates represent the maximal hypoxic area reached each year, during the sampled periods. Based on the available data, the spatial extent does not appear to vary seasonally.*",
  and by modifying the caption of Fig. 2 so it reads as follows:
  "*Maps showing a compilation of all dissolved oxygen (DO) samples collected every year since 1995, for every station sampled deeper than 200 m. A map may contain data from multiple surveys carried out throughout that year. [...]*".
- We justify this by noting that available data show no clear seasonality and that this study focuses on yearly variations, by adding the following at L102:
  "*The seasonality in air temperature and river runoff do not affect the bottom-water properties of the Laurentian Channel, as they are isolated from the surface by the intermediate layer (CIL). Only the spring bloom and the delivery of autochthonous organic matter could affect the bottom-water oxygen levels on a seasonal scale, but available data show no consistent seasonality.*"

We also substituted Fig. 3 for a version that aggregates the data per season instead of yearly (see below). We could also include a new inset showing a histogram of the number of stations sampled every month from 1995 to 2021.

[Figure]

In addition to the changes related to sampling time, we modified Figure 2 to better represent the sampling depths. The figure shows the lowest sampled dissolved oxygen (DO) concentration, for any station for samples deeper than 200 m. Hence, at some stations, the measurement shown might have been sampled above the hypoxic layer. To avoid confusion, we now use different symbols to identify samples shallower than 250 m, and thus highlight that the non-hypoxic waters (DO > 62.5 µM) are found at shallower depths. For example, in the 2009 map shown below, most of the blue stations in the estuary are diamonds, which identify samples recovered between 200 and 250m, while circles show samples collected below 250 m.

[Figure]

To better explain the data aggregation and to include this new information, we modified the Fig. 2 caption to the following:
"*Maps showing a compilation of all dissolved oxygen (DO) samples collected every year since 1995, for every station sampled deeper than 200 m. A map may contain data from multiple surveys carried out throughout that year. The color indicates the lowest DO concentration sampled over the water column, with hypoxic waters in magenta. The symbols indicate if the deepest sample is located*

*between 200 and 250 m (diamond) or below 250 m (circle). The thin black lines delineates the 275 m isobath.*"

**RC2: Line 132: Perhaps replace "alarmingly fast" with "rapidly"? The current language reads a little awkwardly and is somewhat sensational.**

We tamed down the vocabulary and substituted "alarmingly fast" by "rapidly" in the revised manuscript. Irrespective, after more than 20 years of near status quo, the sudden decrease in minimum dissolved oxygen (DO) concentrations is alarming. According to more recent measurements (September 2022), minimum DO may have dropped further to 27-30 µM.

**RC2: Line 134: Is strikes me that you could argue that temperature rose rapidly in the past two years (as oxygen declined rapidly), but you really only speak to the progressive longer term trend here. This comment is also relevant in the conclusions on line 216.**

This is a good point. As the minimum DO concentrations decreased rapidly, the water temperature also increased quickly, from 5.6 to 6.2°C in the same parcel of water, between 2018 and 2021, reaching nearly 7°C in 2020 in the western Gulf of St. Lawrence. We added this information in the revised manuscript at line 153:
"*Concurrently, bottom-water temperatures in the LSLE and the GSL have increased progressively from ~3°C in the 1930's to nearly 7°C in 2020, **including a rapid 1°C increase from 2019 to 2020***."
At L216, we changed "*bottom-water temperatures have increased steadily*" to "*bottom-water temperatures increased by one degree over the same period*".

**RC2: Figure 4: What does the grey area in top panel represent?**

The grey band is a visual guide for the temporal trend of the minimum bottom-water oxygen concentrations in the Lower St. Lawrence Estuary over the period of available field measurements. Its slope is found from a linear least-squares fit  to the data. We added this information to the caption: *"The grey bands indicate the trend, based on a linear least-squares fit  to the data, with the R² indicated on the figure. This figure offers a crude estimate of the trend in the  bottom-water properties, given that the exact sampling depth, although always below 250m, might have varied over the decades."*
We also add a similar band to the temperature data.

**RC2: Figure 5: Can you specify when these samples were taken each year? Is it possible that they fluctuate enough over the season that a snapshot might miss some intra-seasonal variations? If so, this would be a relevant discussion point to communicate some of the uncertainty in the study.**

Like Figure 2, this figure shows an aggregation of all samples taken in any given year. We modified the caption to indicate the periods over which the data presented in Figure 5 were acquired (added text in bold): "*Transects of **all dissolved** oxygen (DO) concentration **measurements** along the Laurentian Channel, **sampled in spring, summer and fall** 2003, 2009, 2017 and **in summer and fall** 2021, from the head of the channel (left) to Cabot Strait (right).*"
Most of the surveys were conducted in spring and summer, winter surveys were only conducted on three consecutive years in recent years (2018, 2019, 2020), and the data were aggregated per year in Figs 2, 3 and 5. Nevertheless, beyond the mean annual variations, no

distinct seasonal variations could be discerned. This is not unexpected because the deep, hypoxic waters, are isolated from the atmosphere by the Cold Intermediate Layer. Only the spring bloom and the delivery of autochthonous organic matter to the seafloor could affect deep oxygen levels on a seasonal scale, but the available data show no consistent seasonality (see figure below in which each colored star represents a given year). We specify this by adding, at L102:

"*The seasonality in air temperature and river runoff do not affect the bottom-water properties of the Laurentian Channel, as they are isolated from the surface by the intermediate layer (CIL). Only the spring bloom and the delivery of autochthonous organic matter could affect the bottom-water oxygen levels on a seasonal scale, but available data show no consistent seasonality.*"

We modified L69-70 from "*During most of the ice-free season, the water column of the LSLE can be described as a three-layer system on the basis of its thermal stratification.*" to "*The LSLE is a strongly stratified system, and can be described as a three-layer system.*", and we added the bold text at L74: "*bottom layer (>150 m deep) flows sluggishly landward, **isolated from the atmosphere** (Dickie & Trites, 1983), for 4 to 7 years from the continental shelf-break to the head of the Laurentian Channel (Bugden et al., 1991; Gilbert, 2004).*".

As mentioned above, we now also explicitly indicate that the data are aggregated per year, and justify this approach by noting that available data show no clear seasonality and that this study focuses on yearly variations. To further justify our assumption, we substituted Fig. 3 for a version in which the data are aggregated monthly instead of yearly.

[Figure]

**RC2: Figure 6: Is there an explanation why there is 20-50% LCW fraction in the ~275 m water between 200 and 500 km? The only reference to this figure is that LCW is now almost null, but this seemingly residual or preserved LCW fraction was not discussed, and I think it should be.**

Since the bottom waters require 4-7 years to transit from the mouth of the Laurentian Channel to its head (at the head of the LSLE), the layer of 20-50% LCW that appears at ~250m depth at the landward part of the transect is likely bottom water that is several years older and richer in LCW than waters now entering the Gulf at Cabot Strait. We address this by adding the following in section 3.4, at L 242:

"*As the waters require 4 to 7 years to transit from the continental shelf break to the head of the Laurentian Channel, the presence of 20-50% LCW in the bottom waters close to the head of the channel (leftmost to central portion of Figure 7) reflects the proportion of LCW that entered the channel several years ago. The presence of residual LCW at the head of the LC suggests that*

*bottom-water dissolved oxygen (DO) concentrations will further decline when the nearly undiluted oxygen-poor NACW reach the head of the LC.*"

We also specify more clearly where the uncertainty is coming from by adding the text in bold at L240: "*within the uncertainty (5%)* **of the eOMP method (see Jutras et al., 2020b)**."

The null LCW fraction seen above and below the LCW-enriched layer is due to the presence of the overlying CIL and underlying NACW. When the presence of the LCW, distinct for the NACW, was first resolved in Jutras et al. (2020b), the analysis revealed that the LCW and NACW that make up the bottom waters of the Laurentian Channel are not perfectly mixed. A layer of water richer in LCW sits on top of a NACW-rich layer in the Lower St. Lawrence Estuary (LSLE).

**RC2: Line 160: Are there data to report here about organic matter and nutrient exports? Minimally it would seem relevant to report a quantitative change, or maximally a figure of the long-term data.**

A limited historical record (1995-2012) of organic matter and nutrient exports from the St. Lawrence River to the Upper St. Lawrence Estuary was published by Hudon et al. (1997), but no clear trend could be identified over this period. Unfortunately, we could not find a multi-decadal time series of nutrient loading for the St. Lawrence River, as is available for the Mississippi River (Turner et al., 2007). The Mississippi River record reveals a large anthropogenic increase in nutrient loading since about 1970. Given the geographic proximity of the two drainage basins, and the similar agricultural practices and industrial activities, it would be reasonable to assume that nutrient loading to the St. Lawrence River has also increased over the last decades. Indirect accounts of nutrient loading certainly support this hypothesis, as do the increase of nitrogen and phosphate fertilizer sales in Quebec and Ontario (Statistics Canada, 2018), as well as the, respectively 3.8 to 4.5-fold, increase in the estimated net anthropogenic P and N inputs to watersheds of the St. Lawrence Basin over the last century (Goyette et al., 2016). The impact of nutrient loading to the Lower St. Lawrence Estuary (LSLE) was addressed in detail in Jutras et al. (2020a) in which the authors present a three-layer box model and budget of nutrients in the LSLE. In this paper, they conclude that the majority of nutrients delivered to the LSLE originate from upwelling of nutrient-rich waters at the head of the channel. Whereas bottom-water nutrient concentrations have increased noticeably since at least the early 1960's, they reached near steady-state values in the 1990's. Instead of simply citing these references at the end of L161, we detail these findings in the revised manuscript as follows:

"*Whereas a multi-decadal time series of nutrient discharge from the river to the estuary is not available, a record of fertilizer sales within the drainage basin is. The sales of phosphorus-based fertilizers grew rapidly in Quebec and Ontario starting in the 1940s and peaked in the late 1980s, after which they decreased slightly in the 1990's and came back to 1980 levels throughout the new millennium. The sales of nitrogen-based fertilizers increased steadily from the 1940s to 1980, stabilized throughout the 1980's and 1990's, but have since continued to increase significantly (Statistics Canada, 2018). Goyette et al. (2016) estimated that the net anthropogenic P and N inputs to watersheds of the St. Lawrence Basin have increased by respectively 3.8 and 4.5-fold over the last century.*"

**RC2: Line 185-190: I appreciate that the methodology has been presented before (Jutras et al. 2020b) to partition the delta DO among driving factors, but I think a brief explanation of the approaches here would help make this paper stand alone.**

We feel that it would be unreasonable to repeat details of this methodology here as it would unduly lengthen the manuscript. The main objective of the present paper is to present the temporal variations in the spatial extent of the hypoxic zone. The other objective is to highlight the sudden drop in dissolved oxygen (DO) concentrations since publication of the Jutras et al. (2020b), in which the most recent data are from 2018. The identification of the mechanisms responsible for this sudden drop in DO does not represent a new scientific contribution, since it is the same as for the 2008-2018 period covered in Jutras et al, (2020b). Hence, no new information would come from further discussion on these causes. To make the objective of the paper clearer, we reorganized the Result section of the paper. First, we moved section 3.4 (Areal extent of the hypoxic zone) to section 3.2. Second, we combined sections 3.2 'Causes' and section 3.3 'Sources of water' into a single section, removing some repetitions and making it clearer what the novel contributions presented in this paper are by adding, at L236:
"*Here, we apply the eOMP method used in Jutras et al. (2020b) (Section 2) to reconstruct the causes of the 2019 to 2021 oxygen decline (Figure 8).*"

We still added more details on the eOMP method, in the Method section, by adding the text in bold at L107:
"***In this method, a set of linear equations that describe the properties ($S_P$, $\delta^{18}O(H_2O)$, temperature, alkalinity, oxygen and nutrient concentrations) of a parcel of water is used to determine*** *the relative contributions of the different water types* ***that make up that parcel of water, given a definition of these water types in terms of the available water properties. Unlike the T-S diagram method, the eOMP method accounts for diapycnal mixing and provides estimates of biogeochemical changes that occurred between the water type formation and the measurement locations. Details of the application of this method to the current data set can be found in Jutras et al. (2020b).***"
and, in the Results section 3.4 at L249:
"*Another 30% of the 2018 to 2021 deoxygenation is attributed to the increased biological oxygen consumption rates during the transit of waters along the Laurentian Channel in response to the temperature increase (Figure 8),* ***estimated from the empirical temperature dependence of bacterial respiration rate $Q_{10}$ (Jutras et al., 2020b; Genovesi et al., 2011; Bailey & Ollis, 1986)****. The remaining 10% decrease is due to an* ***increased dissolved oxygen consumption in the North Atlantic****, also likely due to higher water temperatures,* ***as estimated from the results of the eOMP analysis at the entrance of the Laurentian Channel****.*"
We also give more details about the parent water masses properties at L209:
"*The former are colder and carry more DO than the latter (-0.7°C to 3.2°C and 310 to 280 µmol/kg of DO for the LCW, compared to 4.4°C to 17.7°C and 155 to 250 µmol/kg of DO for the NACW: Jutras et al., 2020b).*"
and we make it clear from the beginning of section 3.3 (Causes) that we do not introduce new information on the causes of deoxgenation, by adding, at L190:
"*The causes of deoxygenation in the Lower St. Lawrence Estuary from the 1930s to 2018 were discussed in detail elsewhere (e.g. Jutras et al., 2020b; Gilbert et al., 2005) and, thus, we here summarize their contribution to the most recent decline of minimum dissolved oxygen (DO) concentrations (2018 to 2021).*"
Finally, for more details about how the water masses mix, we refer the reader to Jutras et al. (2020b) at L236, since these topics have already been covered in that paper:
"*Here, we apply the eOMP method used in Jutras et al. (2020b) (Section 2) to reconstruct the causes of the 2019 to 2021 oxygen decline (Figure 8). We refer the reader to the original paper for details on the properties of the different water masses, their distribution along the Laurentian Channel, and the temporal variability in their relative contributions to the bottom waters of the Laurentian Channel.*"

**RC2: Figure 8: can you express what error bars represent here? How were they calculated?**

The error bars represent the uncertainty associated with the calculation of the areal extent of the hypoxic zone, as described at L110-111. The uncertainty is, in great part, a consequence of the low spatial resolution sampling, and more specifically the extrapolation of the area between the last hypoxic station and the first non-hypoxic station along the Laurentian Channel. We added a reference to L110-111 in the revised caption, as follows:
"*The error bars indicate the uncertainty associated with each year's estimate, based on the spatial sampling rate (see section 2).*"